

# Ice sheet and palaeoclimate controls on drainage network evolution: an example from Dogger Bank, North Sea

Andy R. Emery[1], David M. Hodgson[1], Natasha L.M. Barlow[1], Jonathan L. Carrivick[3], Carol J. Cotterill[2], Janet Richardson[1], Ruza Ivanovic[1], Claire Mellett[4]

[1]School of Earth and Environment, University of Leeds, UK
[2]British Geological Survey, The Lyell Centre, Edinburgh, UK
[3]School of Geography, University of Leeds, UK
[4]Wessex Archaeology, Salisbury, UK

*Correspondence to:* Andy R. Emery (ee06ae@leeds.ac.uk)

**Abstract.** Submerged landscapes on continental shelves archive drainage networks formed during periods of sea-level lowstand. The evolution of these postglacial drainage networks also reveals how past climate changes affected the landscape. Ice-marginal and paraglacial drainage networks on low-relief topography are susceptible to reorganisation of water supply, forced by ice-marginal rearrangement, precipitation and temperature variations, and marine inundation. A rare geological archive of climate-driven landscape evolution during the transition from ice-marginal (c. 23 ka BP) to a fully submerged

marine environment (c. 8 ka BP) is preserved at Dogger Bank, in the southern North Sea.

In this study, our analysis of high-resolution seismic reflection and Cone Penetration Test data reveal a channel network over a 1330 km$^2$ area that incised glacial and proglacial lake-fill sediments. The channel network sits below coastal and shallow marine sediments, and is therefore interpreted to represent terrestrial drainage network. When mapped out, the channel form morphology reveals two distinct sets. The first set comprise two low sinuosity, wide (> 400 m) channels that contain

macroforms of braid and side bars. These channels are interpreted to have originated as proglacial rivers, which drained the ice-sheet margin to the north. The second set of channels (75-200 m wide, with one larger, ~400 m wide) have higher sinuosity and form a sub-dendritic network of tributaries to the proglacial channels.

The timing of channel formation lacks chronostratigraphic control. However, the proglacial rivers must have formed as the ice sheet was still on Dogger Bank, before 23 ka BP, to supply meltwater to the rivers. Ice-sheet retreat from Dogger Bank

led to reorganisation of meltwater drainage and abandonment of the proglacial rivers. Palaeoclimate simulations show a cold and dry period at Dogger Bank between 23 and 17 ka BP. After 17 ka BP, precipitation increased, and drainage of precipitation formed the second set of channels. The second set of rivers remained active until marine transgression of Dogger Bank at c. 8 ka BP. Overall, this study provides a detailed insight into the evolution of river networks across Dogger Bank, and highlights the interplay between external (climate) and internal (local) forcings in drainage network evolution.



## 1 Introduction

Postglacial drainage patterns in the North Sea have become a focus of interest in recent years since the growth in archaeological exploration of the submerged landscapes of the Northwest European Continental Shelf (Bailey et al., 2017; Coles, 1998; Flemming et al., 2017). The adoption of seismic reflection data acquired for oil and gas exploration by the archaeological community has allowed mapping of extensive terrestrial drainage networks throughout the Southern North

Sea (Fitch et al., 2005; Gaffney et al., 2007, 2009; Hepp et al., 2017, 2019; van Heteren et al., 2014; Prins and Andresen, 2019; Tappin et al., 2011). These subsurface mapping projects focussed on rivers as they are likely sites of human occupation. Core and sediment-based palaeoenvironmental research has augmented seismic mapping studies (Brown et al., 2018; Gearey et al., 2017; Tappin et al., 2011), and put human-landscape interaction into the wider context of Late Quaternary landscape evolution of the North Sea during a period of changing climate (Bicket et al., 2016; Bicket and

Tizzard, 2015; Phillips et al., 2017; Tizzard et al., 2014).

Previous exploration of submerged landscapes have used low-resolution 2D or 3D seismic reflection surveys designed to target deeper oil and gas reservoirs (Fitch et al., 2005; Gaffney et al., 2007, 2009), or combine oil and gas datasets with sparse high-resolution 2D seismic reflection data (Coughlan et al., 2018; Hepp et al., 2017, 2019; Prins and Andresen, 2019). Whilst this enables drainage networks to be identified, there is little information to constrain sedimentary and geomorphic

processes, and therefore the controls on landscape evolution. The availability of new, high-resolution datasets from windfarm site investigation allows more detailed investigation of shallower submerged landscapes (Cotterill et al., 2012, 2017a). Dogger Bank is covered by a large (1500 km$^2$), 2D seismic reflection data grid and geotechnical logs acquired as site investigation for the Forewind windfarm projects.

The evolution of the terrestrial landscape of Dogger Bank over 10 kyr timescales, during a period of marked climate and far-

field base-level change, has mainly focused on glacial (Cotterill et al., 2017b; Emery et al., 2019a; Phillips et al., 2018) and coastal stratigraphy (Emery et al., 2019a). Prins and Andresen (2019) established a transition from subglacial channel to terrestrial drainage in a study area 150 km northeast of Dogger Bank, but detail of the terrestrial landscape evolution at Dogger Bank during and after ice-sheet retreat has yet to be established. Furthermore, the link between external, climatic changes at Dogger Bank and the internal processes of drainage reorganisation, such as river piracy (Bishop, 1995; Shugar et

al., 2017) and landscape evolution are explored for the first time in this study with the integration of palaeoclimate model simulations.

In this study, our aim is to describe in detail the timing and processes of formation of channel networks observed in the seismic reflection data using stratigraphic relationships, alongside palaeoclimate model simulations, to identify changes in temperature and precipitation at Dogger Bank. We identify, for the first time, the evolution of a low-gradient terrestrial

drainage network on the low-relief topography at Dogger Bank under changing climate and global mean sea-level rise. We explore the role of ice-sheet meltwater and subsequent precipitation changes in forming a well-developed channel network on a land surface with low topographic relief. This regional picture of changing drainage patterns in the North Sea during the





Late Pleistocene and Holocene has implications for human populations and migration during this period of climatic warming and global mean sea-level rise.

## 2 Setting

Dogger Bank, in the southern North Sea (Figure 1), is a present-day bathymetric high (15-30 m below Mean Sea Level, MSL) surrounded by deeper water (> 50 m below MSL). Site investigations for windfarm construction on Dogger Bank (Figure 1) has provided a wealth of high-resolution 2D seismic reflection data and geotechnical boreholes, split into two Tranches, A and B. Tranche B is the focus of this study (Figure 1). Dogger Bank comprises a stratigraphically-complex archive of environmental change from at least the Last Interglacial (Marine Isotope Stage (MIS) 5e, c. 125 ka BP) through to the present day (Cameron et al., 1987, 1992; Cotterill et al., 2017b; Gibbard et al., 1991). The onset of glaciation at Dogger Bank was likely to have been during MIS 3, around 30 ka BP (Carr et al., 2006; Clark et al., 2012; Hughes et al., 2016; Phillips et al., 2017; Roberts et al., 2018; Sejrup et al., 2000), with Dogger Bank fully deglaciated by 23 ka BP (Emery et al., 2019a; Roberts et al., 2018). A large proglacial lake was present during deglaciation, which initially filled with sediment and then was subaerially exposed, along with thrust-block moraine complexes and outwash fans, after the ice retreated (Cotterill et al., 2017b; Emery et al., 2019a) (Figure 1). Channels incised into the glaciogenic sediments formed during this period of subaerial exposure (Cotterill et al., 2017b). These channels form part of the extensive networks of post-LGM North Sea channel-fills mapped by the North Sea Palaeolandscapes Project (Fitch et al., 2005; Gaffney et al., 2007, 2009) and other studies (Cameron et al., 1987; Coughlan et al., 2018; Hepp et al., 2017, 2019; van Heteren et al., 2014; Hijma et al., 2012; Prins and Andresen, 2019; Salomonsen, 1993). The channel network was buried during subsequent marine transgression around 10 ka BP, with complete inundation occurring around 8 ka BP (Brooks et al., 2011; Emery et al., 2019b; Shennan et al., 2000; Sturt et al., 2013). Shallow marine sand, varying in thickness from 0 m up to 25 m, was then deposited during the Late Holocene (Cotterill et al., 2017b; Emery et al., 2019b).The stratigraphy provides an archive of the transition from MIS 5 marine sediments, through to glacial and terrestrial environments, followed by a return to marine conditions during MIS 1, 11.7 ka BP to present (Cotterill et al., 2017b, 2017a; Emery et al., 2019a, 2019b; Phillips et al., 2018). These new high-resolution data have helped to constrain the timing and extent of glaciation and subsequent landscape evolution at Dogger Bank. From oldest to youngest, the lithostratigraphic formations encountered in the study area belong to the Dogger Bank Formation (glaciotectonised and glacial outwash sediments), the Botney Cut Formation (proglacial lake-fill sediments, an unnamed formation (the channel-fills), and the Nieuw Zeeland Gronden Formation (shallow marine sand) (Cotterill et al., 2017b; Emery et al., 2019a; Stoker et al., 2011).





## 3 Methods

A large, integrated sub-surface dataset of 2D seismic reflection profiles and geotechnical logs acquired for site investigation of Tranche B of the Forewind windfarm project (Figure 1) was used in this study.

### 3.1 Seismic reflection data and interpretation

A dense, 2D grid of shallow seismic reflection data was available to this study, totalling 17,000 line km in Tranche B (Cotterill et al., 2017b); 629 NE-SW oriented lines (mainlines) are spaced at 100 m intervals and 75 NW-SE oriented lines (crosslines) are spaced at 500-1000 m intervals. Two ships were used, employing the same 1.6 kJ sparker source. Data were recorded in StrataView, then imported to ProMAX for processing, where a bandpass filter with a 100 Hz lowcut and an 800 Hz highcut was applied, then exported to SEG-Y. The sparker source has a maximum vertical resolution of approximately

1.25 ms, which gives a vertical resolution of ~1 m in the shallow section. Reflections are resolvable until 180 ms (~150 m), below which the signal becomes too weak to resolve.

Seismic reflection data were interpreted using IHS Kingdom Suite. Maps were generated from interpreting seismic grid lines in two-way time (TWT). P-wave velocities derived from local geotechnical data were used to convert TWT to depth. Sea-water velocity was taken to be 1505 m/s, and a sediment velocity of 1600 m/s was used (Cotterill et al., 2017a). Seismic

horizons were gridded to maps with a 10 m grid square using the flex gridding algorithm in Kingdom Suite, then exported to QGIS for further processing.

Seismic interpretation was undertaken by identifying distinct seismic facies and major bounding surfaces between them. Seismic facies were identified and named based on Mitchum et al. (1977) and extended for interpretation of glacial sediments by Emery et al. (2019b). Seismic facies were correlated to sedimentary facies interpreted from geotechnical logs

to establish a seismic stratigraphic framework for the study area. This framework was used to identify the transition from glacial to terrestrial to marine and map the major bounding surfaces between the different sedimentary environments.

### 3.2 Geotechnical logs

Eighty-three Cone Penetration Tests (CPTs), up to 50 m below seabed, were acquired throughout Tranche B (Cotterill et al., 2017b). These tests provide cone resistance measurements that were used as a proxy to grain size through the sediments.

CPTs were used to calibrate sedimentary information to seismic facies observed in the seismic reflection data to constrain sedimentary environment. CPT depth was converted to TWT by using the sediment velocity of 1600 m/s.

### 3.3 Geomorphic interpretation

Channel networks were digitised from the seismic horizon interpretation to polygon shapefiles in QGIS by mapping channel-forms where underlying reflectors are truncated, using this method the upstream contributing area of the catchment was not

needed to extract the stream network. Each channel or channel network was ascribed an individual shapefile. Centrelines for



each of the channels were digitised as line shapefiles. The shapefiles were projected in UTM 31N. Centreline shapefiles were then imported into a python script (Grieve, 2020) that measured centreline length and the straight-line distance from the shapefile start to end from UTM coordinates. These two lengths were used to calculate sinuosity. Centreline shapefiles were also used to extract the channel base long profiles from the depth-converted seismic horizons. As the seismic horizon was

gridded at 10 m, long profile points were automatically extracted by the QGIS Profile Tool plugin every 10 m.

## 3.4 Palaeoclimate modelling

Palaeoclimate simulations provide an estimate of changing climate at Dogger Bank since the Last Glacial Maximum. These equilibrium-type simulations were run with the coupled ocean-atmosphere-vegetation general circulation model, the Hadley Centre Coupled Model version 3 (HadCM3; Gordon et al., 2000; Pope et al., 2000; Valdes et al., 2017). They broadly follow

the protocol described by Ivanovic et al. (2016), opting for a melt-uniform freshwater scenario that conserves water across the deglaciation in accordance with the prescribed ice sheets. Two sets of simulations were analysed; one uses the global ICE-6G_C ice sheet reconstruction (Peltier et al., 2015) performed at 1000 yr intervals 26-21 ka BP and 500 yr intervals from 21-0 ka BP (the same simulations are described in more detail by Morris et al. (2018) in their Supplementary Materials and Methods), and the other uses the GLAC-1D global ice-sheet reconstruction (Briggs et al., 2014; Ivanovic et al., 2016;

Tarasov et al., 2012, 2014; Tarasov and Peltier, 2002) at 500 yr intervals from 26 ka BP to present that are otherwise identical to the ICE-6G_C set. The GLAC-1D ice-sheet model uses the DATED-1 chronological database for the Eurasian Ice Sheet (Hughes et al., 2016), which gives a realistic reconstruction of the ice sheet and palaeogeography of the British Isles, and thus provides the more up-to-date chronology for Eurasian ice sheet evolution. However, the DATED-1 database shows Dogger Bank to be glaciated until 19 ka BP, as opposed to deglaciated by 23 ka BP (Emery et al., 2019a; Roberts et

al., 2018). Therefore, the climate evolution simulated in the Dogger Bank area may be biased, and simulated as too young during this early time window (23-19 ka BP). Nevertheless, it should provide a more faithful representation of climate thereafter. From these simulations, we extracted 50-year climate means for annual precipitation, total annual evaporation and annual temperature for the 0.5 x 0.5° model grid square centred at 55°N, 3.75°E, which covers Dogger Bank.

## 4 Results

## 4.1 Seismic stratigraphic interpretation

Three seismic stratigraphic units were established based on previous investigation of the stratigraphic architecture of the study area (Cotterill et al., 2017b; Emery et al., 2019b, 2019a). The basal unit (Basal Seismic Unit) is separated from the younger two units (Channel-fill Unit, Upper Seismic Unit) by a major unconformable surface (Horizon Z) that is mapped across the study area.



### 4.1.1 Basal seismic unit

The basal seismic unit comprises three main seismic facies, with minor contributions of other seismic facies. Generally, the east and southwest of the study area is characterised by high-amplitude, varying frequency reflectors and asymmetric and symmetric serrate, inclined, patchy and sporadic reflectors (See Emery et al. (2019b) for description of terminology), termed sub-unit 1. The central study area is dominated by high-frequency, medium-amplitude parallel reflectors that onlap or drape previous stratigraphy, infilling a depocentre (Figure 1), termed sub-unit 2. The northwest of the study area mainly comprises low-amplitude, low-frequency variable to transparent seismic facies, termed sub-unit 3. The basal seismic unit is bounded above by Horizon Z.

### 4.1.2 Horizon Z – unconformable surface

Horizon Z is present across the study area and truncates the underlying basal seismic unit, and therefore represents an unconformity (Figure 2). Channel forms mantle Horizon Z, and incise the basal seismic unit. Commonly, Horizon Z is coincident with the seabed (Figure 2). The depth to Horizon Z, relative to mean sea level, is shown in Figure 3. In seismic section, Horizon Z is generally identified by a continuous, medium- to high-amplitude reflector, especially where coincident with the seabed. In the north of the study area, Horizon Z loses reflectivity and can only be interpreted by differences in seismic facies above and below. In some areas, Horizon Z is high-amplitude and overlain by a thin unit of further high-amplitude reflectors.

### 4.1.3 Channel-fill seismic unit

The channel-fills above Horizon Z comprise varying seismic facies. Channel forms vary in size and morphology, as described further in Sect. 4.4. Larger channel-fills often cause acoustic blanking of the underlying seismic reflections. Channel-fill architecture is variable. The dominant channel-fill seismic facies comprises high-frequency, high-amplitude, continuous reflectors that are generally subparallel to the base of the channel, or horizontal, which varies in thickness between and along channels (Figure 4 and Figure 5). High-amplitude reflectors can also be mounded externally with parallel to tangential oblique reflectors internally (Figure 4). Typically, above this high-amplitude fill is a low-amplitude to transparent, low-frequency fill with parallel and horizontal, draped, or sigmoidal reflectors (Figure 4 and Figure 5). In some cases, the fill is divergent in the high-amplitude section, and comprise stacked channel-fills (Figure 4). Prograding fill is observed in the low-amplitude, low-frequency section of some of the channel-fill seismic facies (Figure 4). In other channels, the fill pattern is divergent in the basal high-amplitude reflectors, which are overlain by low-amplitude to transparent fills (Figure 5).



### 4.1.4 Top channel-fill horizon

The horizon separating the channel-fill seismic unit from the overlying upper seismic unit is variable. Typically, this horizon
comprises a single medium- to high-amplitude horizontal reflector (Figure 3 and Figure 4) but can also be draped over the
partial channel fill (Figure 4 and Figure 5). Where channels are absent, this horizon is coincident with Horizon Z.

### 4.1.5 Upper seismic unit

The youngest seismic unit is present between the seabed and Horizon Z, and sometimes partially fills the channels (Figure 1,
Figure 3, Figure 4, and Figure 5). This upper seismic unit comprises low-amplitude to transparent seismic facies. In central
and northern parts of the study area, where the upper seismic unit is thickest, low-frequency, low-amplitude, west to
southwest-dipping sigmoidal to tangential oblique and shingled reflectors are present (Figure 2). The upper seismic unit is
absent where Horizon Z is coincident with the seabed.

Two large, elongate features, oriented approximately NNW-SSE, up to 2.5 km wide and 15 km long, are also present in the
centre of the study area that incise into Horizon Z and the basal seismic unit (Figure 2). In the north of the study area, the
largest elongate feature can be observed to incise through the channel-fill unit and into the basal seismic unit, suggesting that
these erosive features are younger than the channel-fill unit.

### 4.2 Geotechnical log interpretation

Ten CPTs intersect the channel-fills of Horizon Z (Figure 3). Low cone resistance correlates to clay, and high cone
resistance correlates to sand (Robertson, 1990). CPT facies of each of these ten geotechnical logs, plus two logs that did not
penetrate channel-fills, were interpreted for each seismic unit by correlating CPTs to the seismic reflection data. This
correlation was made by converting CPT depth to TWT using the sediment velocity of 1600 m/s (Cotterill et al., 2017a).
The upper seismic unit has high cone resistance values, as seen in CPTs H, I, K, M, V, and W (Figure 3), which implies a
sand-rich unit. The basal seismic unit is highly variable depending on which sub-unit is encountered. Sub-unit 1 has mostly
low cone resistance values, suggesting it is clay-rich (CPTs H and O) but can also be interbedded with more sand-rich layers
(CPT N; Figure 3). Sub-unit 2 is dominated by clay-rich sediments with rare thin layers of coarser material (CPTs I, K, P, V,
and W; Figure 3). Sub-unit 3 is variable, but dominated by high cone resistances intercalated with intermediate responses,
implying silty and sandy sediments (CPT R; Figure 3, Robertson, 1990).
The channel-fill signatures differ between each CPT, varying from clay-rich to sand-rich, and are commonly interbedded. In
CPT W, the channel-fill is clay-rich at the top, between 6 m and 14 m, and sand-rich to the base of the channel-fill at 19 m.
This correlates to the difference in seismic facies within the channel-fill seismic unit, where the clay-rich unit correlates to
low-frequency, transparent seismic facies, and the sand-rich unit correlates to high-amplitude, parallel reflectors (Figure 3).
In contrast, CPT V contains no clay-rich layers, solely comprising intermediate cone resistances, implying a silty to sandy
fill. CPTs I, L, M, and P comprise low cone resistance responses, implying clay-rich facies. CPTs H and N are also



dominated by clay, but have irregularly-spaced, 10-40 cm thick, intercalated silts. The general trend within CPT N shows
fining upwards. CPTs K and O are highly variable in cone resistance throughout the channel-fill seismic unit, with a sandy
base, a clay-rich middle, and a sandy top.

## 4.3 Sedimentary environments

The three seismic units and Horizon Z, when correlated to CPT log facies, suggest a transition between three distinct
sedimentary environments.

### 4.3.1 Glacial and proglacial deposits

The wide range of seismic and CPT facies within the basal seismic unit imply a complicated depositional setting. Sub-unit 1
has varying seismic facies that includes serrate, patchy, inclined and sporadic reflectors, and correlates to fine-grained, clay-
rich facies with siltier interbeds and some sand, such as in CPTs H, O, V and W (Figure 3). The nature of the reflectors
implies deformation of this sub-unit, which is interpreted to be glaciotectonic compression of subglacial and glacial outwash
sediments deposited at the margin of an ice sheet (Cotterill et al., 2017b; Emery et al., 2019a; Phillips et al., 2018). The
rhythmic, parallel, high-frequency reflectors within sub-unit 2 correlate to clay-rich CPT facies in CPT I, P, L, V, and W
(Figure 3). The fine-grained, rhythmic deposits suggest deposition in a low-energy environment, and the basin-filling
geometry of the sub-unit supports an interpretation that these are lake-fill sediments. The third sub-unit is characterised by
low-amplitude reflectors whose geometry implies aggradation at low angles. CPT R (Figure 3) has a variable, but generally
high cone resistance, implying interbedding of sand and siltier units. Sub-unit 3 has varying thickness; it is thicker in the
west of the study area, thinning to a lobate geometry to the east, where it overlies sub-unit 1, and is onlapped by sub-unit 2.
Sub-unit 3 is interpreted to be glacial outwash deposited subaerially in outwash fans during ice-sheet retreat. A full
description of these glacial and proglacial sedimentary environments and their implication for landscape evolution during
ice-sheet advance and retreat is provided in Emery et al. (2019b).

### 4.3.2 Terrestrial and fluvial

Horizon Z is an unconformable surface, into which channel forms have incised. These channel forms vary in length and
width, but form a connected network of channels (Figure 6). There are three main channels, which generally have smaller
tributary channels joining them. The high aspect ratio channel-fills (50:1 up to 100:1) are shallow (up to 15 m deep) but wide
(up to 1 km width). Where Horizon Z is not coincident with the seabed, the channel forms are observed to run parallel to the
centreline of low relief valleys (Figure 3).
Horizon Z is interpreted to represent a composite terrestrial surface that formed after retreat of the ice sheet and infilling of
the proglacial lake (Cotterill et al., 2017b; Emery et al., 2019a). The channels might have originated as a tunnel valley
network. However, tunnel valleys are generally much deeper, with lower aspect ratios (10:1) and highly-undulating thalwegs
(Livingstone and Clark, 2016; Ó Cofaigh, 1996). Furthermore, if the channels were of a tunnel-valley origin, their



stratigraphic position would require a late-stage ice-sheet readvance over the proglacial lake-fill sediments. However, no evidence of deformation related to readvance is recorded within the proglacial lake-fill sediments or anywhere else throughout Tranche B (Emery et al., 2019a). Therefore, we favour a fluvial origin for these channels, which incised into Horizon Z.

The CPT logs have mixed responses within the channel-fill seismic unit, implying different infill histories. Sandy and silty
channel-fill sediments are frequently encountered towards the base of the channels (CPTs K, N, O, V, and W), implying a moderate to high energy sedimentary environment. Fining upwards is also apparent in CPTs K, N, and W, which is characteristic of bar deposits in channel-fills. Clay-dominated facies (CPTs H, I, L, M, P, and the upper section of CPT W) suggest a low-energy sedimentary environment. The clay-dominated facies could also represent brackish or marine deposition during marine transgression. Without detailed sedimentary information provided by cores, and
palaeoenvironmental analyses, such as microfossil assemblages, it is not possible to confirm the depositional environment of these clay-rich facies.

### 4.3.3 Shallow marine

The generally low-amplitude to transparent seismic facies of the upper seismic unit implies a relatively homogeneous sediment. The CPT logs that correlate to the upper seismic unit, such as CPTs V and W, have high cone resistance values
(~30-50 MPa), suggesting the unit is sand-rich. The homogeneous sand and generally eastward-dipping sigmoidal reflectors are interpreted to represent progradation of sand in a shallow marine depositional environment across Horizon Z after marine transgression. This interpretation is further corroborated by shallow marine sands recovered from vibrocores in the southeast of this study area, such as vibrocore 213 (Figure 2), as shown in Emery et al., (2019a).

### 4.4 Geomorphology

Three main channel-fills are identified above Horizon Z, whose widths are greater than 400 m and up to 1000 m wide and 15 m deep (Figure 6). Main channel-fill 1 runs from west to east and is located in the east of the study area. Main channel-fill 2 runs from north to south in the centre of the study area. Main channel-fill 3 runs from northwest to southeast in the west of the study area (Figure 6). A tributive network of smaller channel-fills associated with the large channel-fills also exists, whose widths are up to 250 m, and depths up to 10 m. Longer, isolated channels of a similar scale are also observed within
the study area.

Two forms of channel cross section are observed. The first form corresponds to main channel-fills 1 and 2, with a wide channel incision that comprises numerous smaller erosion surfaces separated by shallower and generally horizontal sections at its base (Figure 4). The second form, corresponding to the tributaries, main channel 3, and the isolated channels, are generally U- or V-shaped with a single deep incision (Figure 5).

The base of main channels 1 and 2 show cross-channel depth variations (Figure 4) from narrow, deep channel sections separated by wider, flat-topped, mounded shallow sections elongated parallel to the channel with internal oblique reflectors





that dip downstream. The deeper sections split and rejoin, with between one and three deep channel sections across the main channel width (Figure 4). The main channels 1 and 2 have low sinuosity (channel 1 = 1.06, channel 2 = 1.05; Figure 7). The tributaries, main channel 3 and isolated channels observed have similar sinuosity (mean of 1.25) and average channel widths

(100 – 150 m; Figure 7), which differ from those of main channels 1 and 2. The tributaries and streams are between two and four times smaller than main channels. There is a large variation in sinuosity of the tributaries, ranging from 1.07 to 1.53, and the isolated channels are straighter, with sinuosity between 1.16 and 1.43. Main channel 3 has a sinuosity of 1.22. The tributaries join the main channels at angles around 90° (Figure 6), implying a perpendicular flow direction to the main channels.

Long profiles of the three main channels and their longest tributaries were drawn from the centre-lines of the deepest point of the channel base (Figure 7). The profiles undulate, but show overall decrease in elevation. Eastward (channel 1), southward (channel 2), and southeastward (channel 3). The tributary channel bases also decrease in elevation and become steeper from the tributary head to the confluence with main channels (Figure 7), implying these channels are cutting down to the main channels. The flow direction of the rivers is interpreted to be the same as the direction of decrease in elevation (Figure 7).

Therefore, the network of channels is a dendritic to sub-dendritic river drainage network (Zernitz, 1932) draining from tributaries into main channels, then out of the study area. The maximum elevation of this drainage network is -32 m, and the minimum elevation is -56 m. Average gradient for the main channels range from 0.2 to 0.9 m/km (0.01° to 0.05°).

## 5 Discussion

### 5.1 Landscape evolution

The northward retreat of the ice sheet from Dogger Bank left a landscape of glaciotectonites, glacial outwash and proglacial lake-fill sediments (Cotterill et al., 2017b; Emery et al., 2019a; Roberts et al., 2018). The resulting landscape surface is likely to have been modified where the seabed and Horizon Z are coincident, and therefore ascertaining the original topography is challenging, though it is likely that any topographic relief was subdued, as areas of this land surface beyond the channels are planar (Figure 3). This is in contrast to the landscape exposed in Tranche A of the Dogger Bank Forewind windfarm project, to the west of this study area, which had an undulating surface of moraine highs and drainage channel lows (Cotterill et al., 2017b; Phillips et al., 2018). During this period of exposure, the land surface would have been a periglacial tundra with limited vegetation (Cotterill et al., 2017b), resulting in desiccation and overconsolidation of the sediments (Cotterill et al., 2017b; Emery et al., 2019a; Mesri and Ali, 1999).

The morphology and low sinuosity of main channels 1 and 2 reflects modern proglacial braided river channels (Carrivick

and Russell, 2013), such as Icelandic glacial outlet rivers, e.g. Jökulsá á Fjöllum (Alho et al., 2005; Bristow and Best, 1993; Carrivick et al., 2007; Maizels, 1989; Marren, 2005; Marren and Toomath, 2014; Vandenberghe, 2001). Braided river channels often form in cold climates, such as proglacial settings, with a high sediment throughput, where there is little vegetation and the channels are unconfined (Bristow and Best, 1993; Marren, 2005). The individual Dogger Bank channels



within the main channel body are separated by shallower sections, interpreted as braided channels separated by mounded

braid bars with internal cross-bedding implying downstream accretion. Given the similarities in morphology to modern systems, we interpret that these channels formed in a proglacial setting, with meltwater containing a high sediment supply from the retreating ice sheet to the north, leading to erosion of tundra-plain surface (Figure 8). However, the width of the braidplains (400-1000 m) is modest and remains constant, which is in contrast to unconfined braidplains from modern day settings, such as Skeiðarársandur, Iceland, which are generally wider (> 1000 m) and distributive.

The location of the proglacial channels was influenced by antecedent topography. The location of Channel 1 parallels a subtle topographic high formed by a tunnel valley-fill overlain by proglacial lake-fill sediments (Figure 9). The tunnel valley-fill has a ~90° bend (from N-S to W-E trend) directly underneath an erosional feature that removes the head of Channel 1. We suggest that Channel 1 also changed trend here from flowing north-south to flowing west-east, to explain why Channel 1 does not reappear beyond the erosional feature (Figure 9). These features eroded to a deeper point than the

base of the channel reaches, implying the channel was removed, as opposed to not being visible below the erosional features. Channel 2 flows down the axis of the former proglacial lake, and is located at the base of a shallow valley (Figure 2, Figure 6). This location implies there was a topographic constraint formed by the top surface of the former proglacial lake-fill sediments, which limited lateral migration and development of a distributive character. The topographic control may have been exacerbated by the clay-rich, overconsolidated, cohesive lake-fill sediments, which would reduce the ability of channels

to migrate laterally.

The braided proglacial rivers must have formed prior to the retreat of the ice sheet down from Dogger Bank. Retreat of the ice down the retrograde, northern slope of Dogger Bank lowered the ice-sheet basal elevation from -60 m to -110 m (Emery et al., 2019a), therefore preventing meltwater from flowing onto and over Dogger Bank. The proglacial rivers formed when the retreating ice sheet was still on the topographic high of Dogger Bank, but after its retreat off the topographic high, the

meltwater and sediment supply would have been insufficient to form rivers of this size and type. The ice sheet retreated from Dogger Bank prior to 23 ka BP (Emery et al., 2019a; Roberts et al., 2018), implying that the lake filled and the proglacial river channels developed prior to this date (Figure 8, Figure 11). The ice sheet retreating northwards down the retrograde slope would have resulted in drainage capture by the northern slope of Dogger Bank. A similar situation of drainage reorganisation due to glacial retreat has been observed in the present day over decadal timescales (Bishop, 1995; Carter et

al., 2013; Shugar et al., 2017). Meltwater would have rerouted parallel to the ice sheet margin and along the northern slope of Dogger Bank, resulting in ponding of meltwater in a ribbon lake to the north of Dogger Bank, supported by observations of proglacial lake-fill sediment accumulation to the north of Dogger Bank (Roberts et al., 2018). The topographic control on drainage and reduction of meltwater supply to the proglacial rivers on Dogger Bank would have resulted in flows that were underfit to the size of the proglacial river channels, or their abandonment in the case of total meltwater switch-off. These

proglacial rivers were therefore likely short-lived (possibly less than 1 kyr) as major conduits, between the proglacial lake filling and ice sheet retreat off the Dogger Bank topographic high. This short lifespan of the river highlights the interplay between ice-sheet retreat and stream reorganisation in controlling the hydrological evolution of a proglacial landscape.



Discharge variability fundamentally controls the geomorphology of river channels (Fielding et al., 2018; Nicholas et al., 2016). The coefficient of variance ($CVQ_p$) of a river system depends on the ratio of annual peak discharge divided by the mean peak discharge. Generally, in modern river systems with a highly variable peak discharges (high $CVQ_p$), macroform structures such as braid bars are not formed or preserved (Amos et al., 2004; Fielding et al., 2009, 2018). In contrast, low $CVQ_p$ rivers readily form and preserve large macroforms with cross-bedding also well preserved. Cross-bedding is rarely seen in the seismic facies as it is likely to be below seismic resolution, but is occasionally present (Figure 4). However, large braid bars are well-preserved and visible in the seismic data (Figure 4), with some evidence of interbedding in CPT logs (Figure 3), and therefore appear to have been stable with limited reworking during the lifespan of the river. This preservation and bedform scale suggests that the river had a steady meltwater supply and low discharge variability, but preservation may have been enhanced by the sudden reduction in discharge when meltwater supply was switched off during northward ice-sheet retreat. It may be expected that rivers experiencing jökulhlaup flood events will have high $CVQ_p$ values compared to annual peak discharge in non-jökulhlaup years (Russell et al., 2006). The steady meltwater supply implies that this sector of the retreating ice sheet did not produce jökulhlaups, whose discharge variability results in geomorphological characteristics similar to those of rivers with high $CVQ_p$ (Carrivick et al., 2004b; Carrivick and Rushmer, 2006, 2009; Guan et al., 2015; Maizels, 1989, 1997; Marren, 2005; Marren et al., 2009; Staines et al., 2015). However, the relationship between $CVQ_p$ and geomorphological characteristic has not yet been tested in rivers experiencing jökulhlaups (Fielding et al., 2018), and remains a topic for future research. There is no evidence within the preserved proglacial river channels or surrounding landscape of glacial outburst floods from the ice-sheet margin at this time (Carrivick et al., 2004a, 2013), which may support the interpretation of macroform and geomorphic preservation being a result of low discharge variability, implying a lack of outburst flood activity. Furthermore, limited aggradation of proglacial sediments contrasts with the transient landscape and associated sediment accumulations of proglacial forelands of jökulhlaup glaciers (Duller et al., 2014). A lack of outburst activity implies a well-ordered, efficient subglacial drainage system, and this is supported by evidence presented by Emery et al. (2019b) from ice streaming and subglacial meltwater channel morphology. It also may explain why there are a lack of Late Weichselian (MIS 2) tunnel valleys in the Dogger Bank area, contrasting with the tunnel valleys formed during previous episodes of ice sheets in this section of North Sea (Cotterill et al., 2017b; Lonergan et al., 2006; Praeg, 2003; Stewart et al., 2013; Stewart and Lonergan, 2011).

The isolated channels and tributaries, and main channel 3, have different morphologies to the two proglacial river channels (1 and 2). The isolated channels are all very similar and are therefore interpreted to have the same origin as tributaries that joined the main channels outside of the study area. The higher sinuosity (Figure 7) of these smaller channels suggests formation under different conditions to the proglacial channels. The direction of drainage of the tributaries and streams is often perpendicular to the flow of the main channels (Figure 6), following pre-existing slopes, such as the valley to the main proglacial channels. These smaller channels also have heads within the study area, unlike the proglacial channels, which suggests that they did not form due to meltwater. The long profiles of these tributaries show them to cut down to the base of the main channels (Figure 7). The sub-dendritic pattern of these smaller channels, combined with their smaller size and



higher sinuosity, and that they steepen into the main channels, suggests they formed later (Figure 8). The increase in sinuosity is interpreted to represent a warmer climate, with a more erodible substrate no longer bound by permafrost. The large, flat areas of high seismic amplitude are interpreted to represent marshy areas with the same seismic character as areas

from which marshy plant macrofossils have been recovered (Wessex Archaeology, 2014). These marshy areas mainly occur over the proglacial lake-fill sediments, implying a low-permeability substrate that would have prevented groundwater flow of rainwater (Figure 8). This in turn led to the development of the sub-dendritic drainage network, which is best developed and preserved over the proglacial lake-fill sediments (Figure 6).

Only the proglacial river channels show evidence for aggradation of sediment within the channels (Figure 4), with little

evidence in the tributaries or overbank deposits (Figure 5). These smaller river channels are only partially infilled by alluvial sediments, with the rest of the infill being shallow marine sand (Figure 5). Models of relative sea-level rise suggest that inundation of the North Sea Basin began around 16 ka BP (Brooks et al., 2011; Kuchar et al., 2012), resulting in a base-level rise for the drainage network (Figure 10), which should result in aggradation within the drainage network. The lack of aggradation may be due to low discharge and sediment flux, with only a small local supply, or due to the drainage network

being distant from the base-level rise, draining into a local depocentre, i.e. the previously-abandoned proglacial river channels.

Marine transgression occurred in the study area between 9.5 and 8.5 ka BP (Cotterill et al., 2017b; Emery et al., 2019b; Shennan et al., 2000), inundating the incisional channel network first (Figure 8). The small size and limited drainage basin area would not have been sufficient for aggradation under a rising base level to outpace the inundation by marine waters.

This may explain the fine-grained channel-fill sediments observed in some CPTs (e.g. CPT L, CPT W; Figure 3), as marine transgression would have modified the sedimentary environment in the sheltered estuaries to low-energy tidal mudflats, as observed elsewhere in the North Sea during Holocene marine transgression (Coughlan et al., 2018; Gaffney et al., 2009; Hepp et al., 2017, 2019; Prins and Andresen, 2019). The final stage of regional landscape evolution was continued marine transgression, with associated ravinement of the pre-existing topography (Figure 3, Figure 8; Cotterill et al., 2017b; Emery et

al., 2019a). The large, elongate features that incise into Horizon Z, the channel-fills, and underlying basal seismic unit, are interpreted to have formed at this stage as large tidal scours. Continued relative sea-level rise and the transport of sediment, shown by the broadly west to east dip direction of sigmoidal to oblique reflectors in the upper seismic unit, resulted in the deposition of shallow marine sand that completed the infill of the channels and the tidal scour features.

## 5.2 Impact of changing palaeoclimate on terrestrial landscape evolution

Between deglaciation of Dogger Bank (> c. 23 ka BP) and marine transgression at c. 8 ka BP, the landscape was subaerially exposed for a 15 kyr period during which the channel network formed. By linking palaeoclimate model data to the stratigraphic observations, it is possible to infer climatic changes that led to the formation of the channels in the absence of age constraints. The overconsolidation of the proglacial lake-fill sediments in the study area has been interpreted as a response to desiccation during subaerial exposure, rather than loading by ice-sheet readvance, as supported by their





stratigraphic position above subglacial and glaciotectonised sediments, and the lack of glaciotectonic deformation within the lake-fill (Cotterill et al., 2017b; Emery et al., 2019a). The desiccation would have required low precipitation, most likely under periglacial conditions. However, the presence of the more sinuous, dendritic channel network incising into the desiccated lake-fill sediments suggests an increase in precipitation. Radiocarbon dates from waterlogged marshland plant remains in boreholes in Tranche B give Bølling-Allerød Interstadial dates (14890-14010 cal BP and 13810-13480 cal BP),

which contain cold-temperate plant pollen such as transitional *Pinus* and dwarf birch *Betula nana* species and sedge, rush and bogbean recovered in the same sample (Wessex Archaeology, 2014). These plant and pollen assemblages are similar to those observed to the east at Slotseng, Denmark (Mortensen et al., 2011), and is in line with observations from records throughout the North Sea Basin (Brown et al., 2018; Gearey et al., 2017; Smith et al., 2007; Tappin et al., 2011). The pollen suggests a transition from arid, periglacial conditions responsible for desiccation of the glacial sediments, to a wetter climate

that allowed formation of marshes and river channels, at some point prior to 15 ka BP.

Effective precipitation (precipitation minus evaporation) trends from the GLAC-1D simulations show a general decrease in precipitation from 26 to 17 ka BP, whereas the ICE-6G_C runs show an increase from 22 ka BP to a maximum at 14 ka BP (Figure 10). The fluctuating, high precipitation outputs from GLAC-1D may be related to the local presence of a modelled ice sheet during this time, when it should have been largely deglaciated. During the same time period, mean annual

temperature (MAT) increased from -12°C at 21 ka BP to 0°C at around 17.5 ka BP (Figure 10), driven by rising atmospheric $CO_2$ and increasing summer insolation. Between 17 ka BP, and marine transgression at c. 8 ka BP, MAT and precipitation continued to increase to 10°C and ~750 mm/yr, respectively. Notably, the large climate excursions documented for the Bølling-Allerød and Younger Dryas periods (e.g. as recorded in the NGRIP ice core record; Figure 10; Andersen et al., 2004) are not captured by either set of simulations due to the nature of the experiment design. These are equilibrium-type

simulations spaced at 500-year intervals and therefore the simulations do not have the temporal resolution to capture abrupt climate events and meltwater pulses from ice melt (such as have been used to simulate these events in the past, e.g. Liu et al., 2009). Nonetheless, the ICE-6G_C model run shows a peak in rainfall during the Bølling-Allerød period, when the North American ice sheet undergoes rapid deglaciation in the ICE-6G_C reconstruction, represented by the removal of large segments of the ice sheet between timesteps in the reconstruction, which is captured by the climate model, affecting surface

energy balance and atmospheric circulation (e.g. Löfverström and Lora, 2017). Important for the Dogger Bank region, the GLAC-1D runs, which have the more accurate local deglaciation history after 19 ka BP (see Sect. 3.4), shows a steady increase in precipitation from 17 ka BP to 11 ka BP, followed by a rapid increase in precipitation during the Early Holocene. We interpret the time period from deglaciation at 23 ka BP, to MAT reaching 0°C and precipitation increase at 17 ka BP, as an arid, periglacial environment. During these 6,000 yrs, desiccation of the glacial sediments occurred, with limited tributive

channel development (Figure 10). After 17 ka BP, rising $CO_2$ and summer insolation along with the strong climatic influence of the deglaciating Northern Hemisphere ice sheets, drives increased precipitation and temperature, which would have resulted in elevated humidly and the onset of ponding and drainage of precipitation on Dogger Bank, as recorded by the





marshy, waterlogged areas in borehole records, and the incision of the channel network and drainage into the previously abandoned large channels over c. 9 kyr (Figure 10).

**5.3 Where did the water go? Palaeogeography of the Southern North Sea**

Numerous studies have identified Late Pleistocene to Holocene channel networks of a similar stratigraphic position to those in this study (Busschers et al., 2007; Fitch et al., 2005; Gaffney et al., 2007, 2009; Hepp et al., 2017, 2019; Hijma and Cohen, 2011; Prins and Andresen, 2019). During the period when main channels 1 and 2 were active as proglacial channels draining the margin of the Eurasian Ice Sheet into the Late Weichselian North Sea Lake, a large proglacial lake that is proposed to have existed to the south of Dogger Bank (Becker et al., 2018; Hjelstuen et al., 2017; Jansen et al., 1979; Murton and Murton, 2012; Roberts et al., 2018; Sejrup et al., 2016; Toucanne et al., 2010). The proglacial channels would have drained directly into this lake (Figure 11), until the ice retreated off the topographic high at c. 23 ka BP to cut off the meltwater and sediment supply. The Late Weichselian North Sea Lake drained rapidly c. 18.7 ka BP through the Elbe Palaeovalley mouth (Becker et al., 2018; Hjelstuen et al., 2017), leaving the Oyster Ground subaerially exposed. After lake drainage, there were two drainage outlets to the ocean: i) the Fleuve Manche system draining south and eastwards (Bourillet et al., 2003; Gibbard et al., 1988; Mellett et al., 2013; Toucanne et al., 2010, 2015), which drained the Rhine-Meuse and Thames, and ii) the Elbe Palaeovalley, which drained the Elbe, Weser and Ems rivers (Figge, 1980; Gibbard et al., 1988; Hepp et al., 2017; Toucanne et al., 2015) into the Norwegian Trough (Figure 11). A third outlet opened after marine transgression inundated the lower-elevation areas north and west of Dogger Bank, eventually inundating the Outer Silver Pit between 12 and 10 ka BP (Brooks et al., 2011; Shennan et al., 2000; Sturt et al., 2013).

The large areas that remain uncovered by similar datasets, especially in relation to the area formerly covered by the Late Weichselian North Sea Lake, makes the location of where the rivers in the study area drained challenging to constrain (Figure 11). Most rivers interpreted from seismic data during the North Sea Palaeolandscapes Project (Fitch et al., 2005; Gaffney et al., 2007, 2009) drain into the Outer Silver Pit Lake (separate to the Late Weichselian North Sea Lake), but it is not known in what direction. The present-day bathymetry of the Oyster Ground shows little topography, with no evidence of transgressed drainage networks expressed at the seabed. The rivers of Dogger Bank may have drained into the Outer Silver Pit Lake, then northwards into the gradually-transgressing northern North Sea via the Wash-Inner Silver Pit and Humber rivers, or southwards into the Fleuve Manche system, or eastwards into the Elbe Palaeovalley (Figure 11). The general direction of palaeoriver flow identified east of Dogger Bank (Hepp et al., 2017, 2019; Prins and Andresen, 2019) is towards the Elbe Palaeovalley, similar to that observed in our study area and the northern area of the North Sea Palaeolandscapes Project, south of Dogger Bank (Fitch et al., 2005; Gaffney et al., 2007, 2009). We propose that the proglacial rivers initially drained into the Late Weichselian North Sea Lake. After the drainage network began to form at c. 17 ka BP, the Elbe Palaeovalley became the mostly likely outlet for the palaeorivers of Dogger Bank (Figure 11). Further investigation of seismic reflection data over a wider area will permit the postglacial stratigraphic evolution of the drainage networks in the





southern North Sea basin to be better constrained, with implications for understanding human interaction and migration through the landscape during the Late Pleistocene.

## 6 Conclusions

Investigation of the high-resolution, integrated dataset of the 2D seismic reflection grid lines and CPT logs has revealed an environment in transition from glacial through terrestrial to marine conditions, marked by Horizon Z, a prominent

unconformity present across the area. Mapping of Horizon Z revealed a network of channels that incise, and therefore postdate, glaciogenic and proglacial lake sediments, but are buried under shallow marine sand. These channels, along with Horizon Z, are interpreted to represent the terrestrial landscape at Dogger Bank that developed during the period between ice sheet retreat and marine transgression.

Two different types and generations of river channels with distinct morphologies have been defined. The first channel set

comprises two, ~400 m wide, low-sinuosity braided rivers. These braided rivers are interpreted to have formed as proglacial meltwater-fed rivers, which drained the margin of the ice sheet prior to its retreat from Dogger Bank at 23 ka BP. Good preservation of macroforms and evidence of cross-bedding may imply that annual mean discharge variability was low, or were preserved either partly or entirely due to sudden abandonment. Potential low discharge variability may suggest a lack of glacial outburst floods, although modern-day analogues have not been studied in terms of discharge variability in rivers

susceptible to outburst floods. Furthermore, there is no evidence of outburst floods for the ice-sheet margin at this time. The second set of river channels are more sinuous and generally narrower (~200 m), form a sub-dendritic network, and cut down perpendicular to the larger river channels, implying they formed later. Palaeoclimate modelling showed a cold, arid period between ice sheet retreat at 23 ka BP and 17 ka BP, when the climate became increasingly warm and wet, which correlates to marsh environments at Dogger Bank c. 14.9 – 13.5 ka BP. The second channel set formed during the time period from 17 ka

BP prior to marine transgression at c. 8 ka BP, during a period of increased precipitation. The first channel set is likely to have drained into the Late Weichselian North Sea Lake, which drained rapidly through the Elbe Palaeovalley at 18.7 ka BP. The second set of channels are likely to have drained through the former Late Weichselian North Sea Lake basin and out through the Elbe Palaeovalley. Overall, this transition from proglacial rivers, to terrestrial drainage with increased precipitation, and the subsequent preservation of the channels, is rarely observed in sedimentary archives, and offers a

valuable insight into the controls of topography and climate on landscape evolution. The evolution of the Late Pleistocene drainage system also provides an opportunity to target submerged sites that can help to improve understanding of how humans interacted with this low relief landscape.





## Acknowledgements

The authors thank the Forewind windfarm project for supply of the data. ARE was funded by the Leeds Anniversary
Research scholarship. Thanks to Bartosz Kurjanski, Prof Brice Rea and Dr Nick Schofield at the University of Aberdeen for
software support. Thanks to Dr Niall Gandy for NetCDF support. We thank Dr. Stuart Grieve for providing the python script
used to calculate sinuosity. Dr Daniel Hepp and Dr Lasse Prins are thanked for their provision of shapefiles of channels in
the North Sea. Prof Nigel Mountney and Prof Colm Ó Cofaigh are thanked for their discussions on an earlier version of this
manuscript. The contribution from RFI was supported by NERC grant NE/K008536/1 and UKRI grant #MR/S016961/1.
CJC publishes with permission of the Director of the British Geological Survey.

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





**Figure 1. Location of the study area and other geographical features of the southern North Sea Basin. Seismic section shows the stratigraphic architecture established in previous studies (Cotterill et al., 2017b; Emery et al., 2019a, 2019b), showing the incised channels observed in this study.**


Earth **Surface**
**Dynamics**
Discussions



**Figure 2. A.** Subcrop map of the major basal seismic unit facies. **B.** Isopach map of the upper seismic unit. Areas of grey hatching are areas where Horizon Z (top basal seismic unit) has been subsequently eroded and/or is coincident with the seabed.







**Figure 3. Depth map of Horizon Z showing locations of CPTs penetrating the channel network. Seismic section (inset) shows the correlation between seismic facies, Horizon Z, and CPTs. CPT logs shown with the seismic unit they correlate to (purples). CPT logs shown with equal scales. Colour bar for CPT logs "lajolla" (Crameri, 2018).**



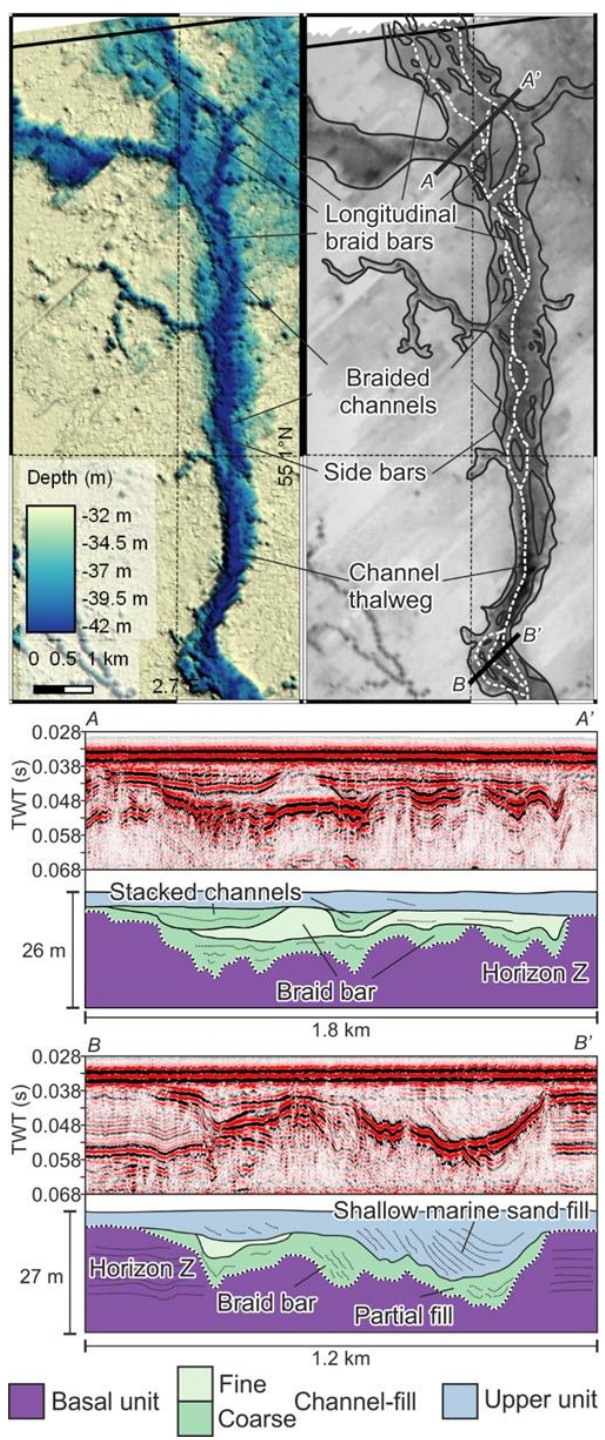

**Figure 4. Detailed map of Horizon Z covering main river channel 2, showing the channel morphology, with braid bars and braided channels. Seismic sections show examples of the vertical channel morphology, with stacked channels and braid bars, and seismic**
**units implying differing sediment fills.**

Earth **Surface**
**Dynamics**
Discussions
EGU

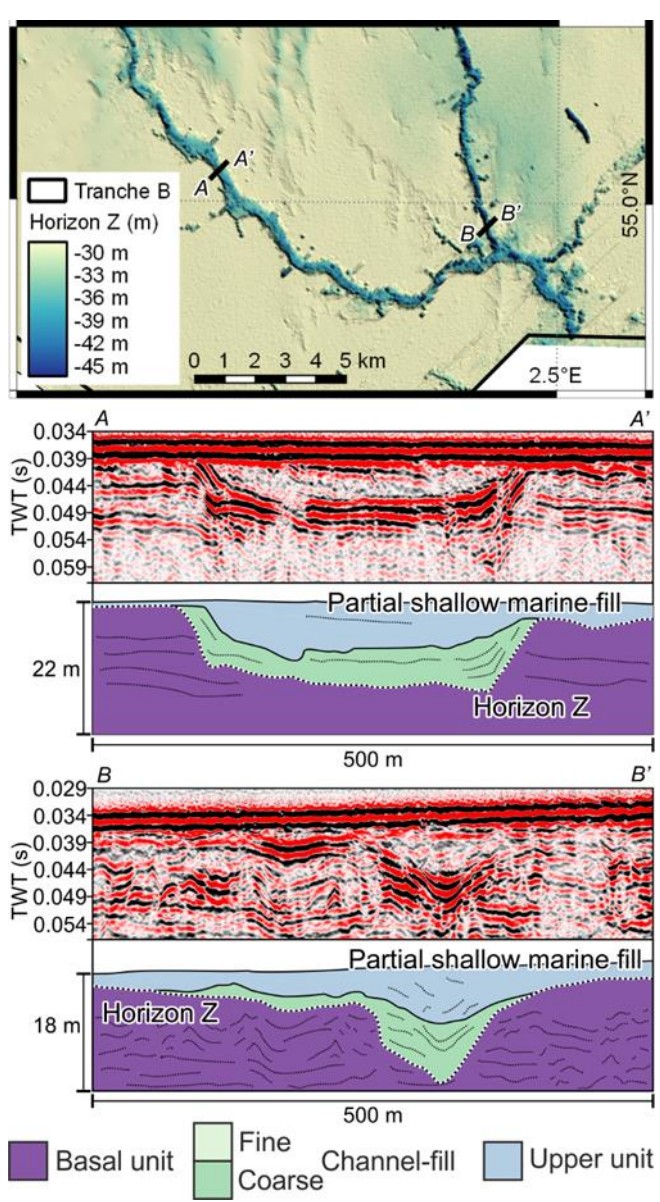

**Figure 5. Detailed map of main river channel 3, showing the more sinuous morphology of the channel and its tributaries. Seismic sections show the more simple, single-channel morphology of the rivers.**





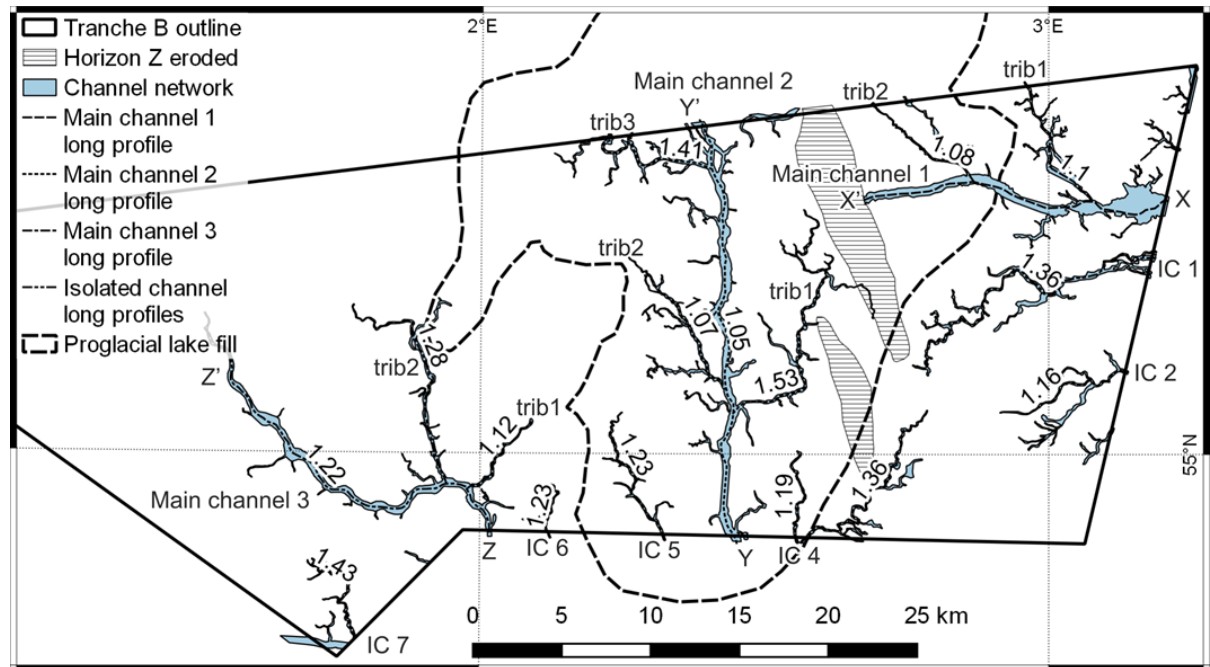

Figure 6. Map of channel network interpreted from Horizon Z showing the three main channels, seven smaller streams, and their tributaries. Numbers correspond to the sinuosity of each individual channel. The outline of the proglacial lake-fill sediment subcrop is shown. IC = isolated channel, trib = tributary.





**Figure 7. Sinuosity and average channel width for the main categories of channels. Main channels 1 and 2 have a distinct morphology when compared to the other channels. Long profiles for channels 1, 2 and 3 show the direction of flow from right to left, and the interaction of tributaries, cutting down and steepening into the main channels. X-X', Y-Y' and Z-Z' are shown on Figure 6.**






Earth **Surface**
**Dynamics**
Discussions

**Figure 8. Conceptual landscape evolution model for the study area, showing a single, representative proglacial channel. 1. Initial drainage of meltwater into the proglacial lake. 2. Proglacial lake gradually infilled with fine, draped sediments. Subsequently, proglacial lake accommodation filled, proglacial river channel incises into the fill. 3. Ice-sheet retreat and drainage reorganisation abandons the proglacial river channels. 4. Temperature and precipitation increase, tributaries incised. 5. Marine transgression floods the river channels first. 6. Final inundation, with wave ravinement, followed by deposition of shallow marine sand.**



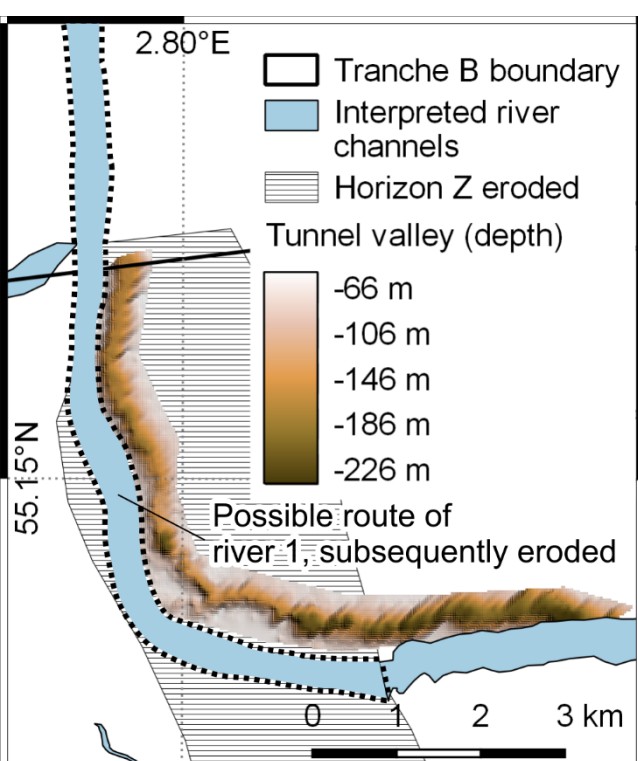

**Figure 9. Tunnel valley observed underneath draped proglacial lake sediments and its possible control on the route of proglacial river channel 1. The proglacial river disappears and cannot be followed under the later erosional feature (grey hatching).**

Earth **Surface**
**Dynamics**
Discussions

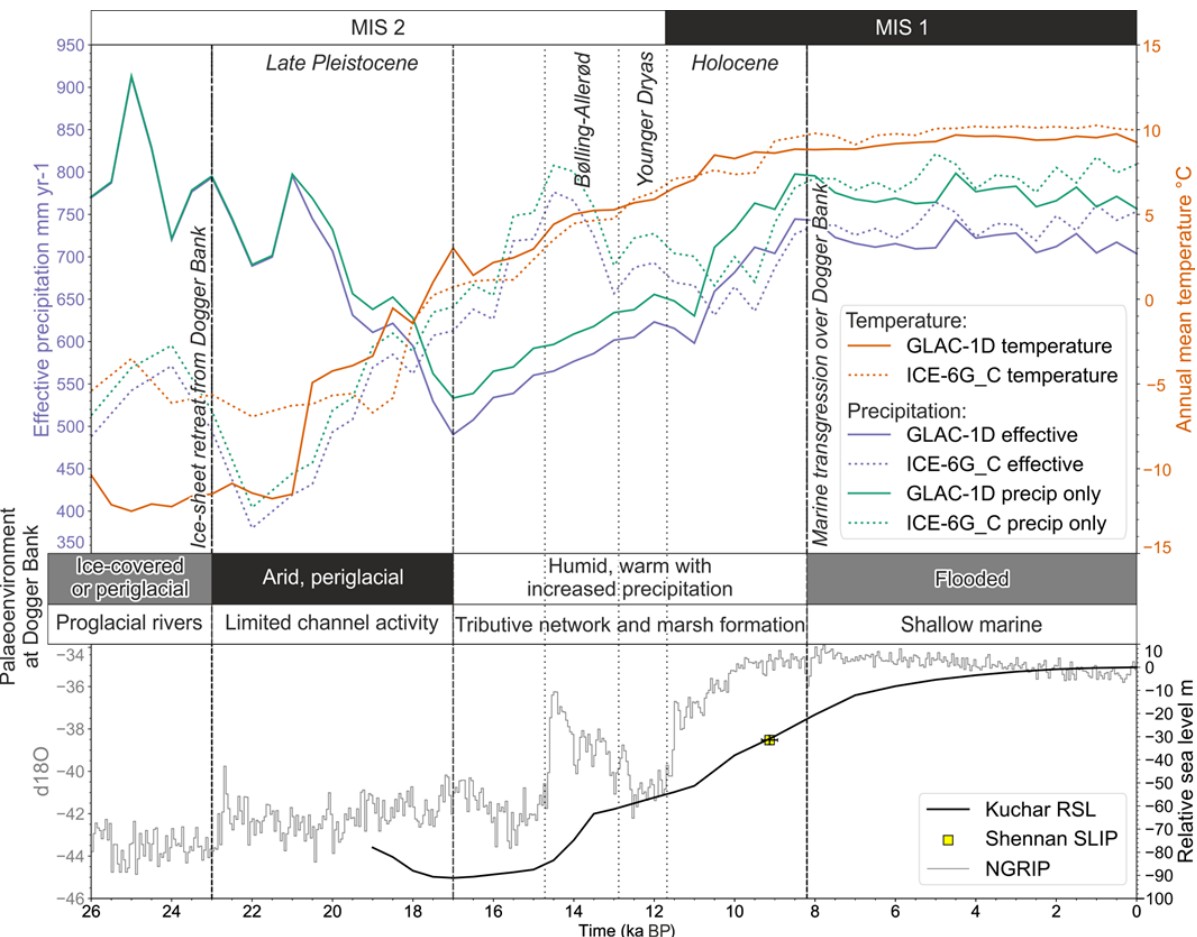

**Figure 10. Palaeoclimate model outputs showing temperature and effective precipitation for GLAC-1D and ICE-6G_C model runs. The interpretation of the palaeoenvironmental conditions at Dogger Bank is a distinct transition from cold and dry to warmer and wetter at approximately 17 ka BP. Also shown is the relative sea-level curve from Kuchar et al., (2012) and the sea-level index point from Shennan et al., (2000), plotted alongside the NGRIP ice core climate record (Andersen et al., 2004). RSL = Relative Sea Level. SLIP = Sea-Level Index Point.**






**Figure 11. Compilation of the Late Pleistocene and Holocene lakes and drainage routes of the terrestrial southern North Sea Basin,**
**along with timings of the existence of the features and the drainage route out of Dogger Bank. The map represents possible**
**drainage routes at different times, where drainage may have been blocked by an ice sheet or ice-sheet retreat opened new drainage**
**routes. The question marks represent the spatial and temporal uncertainty in drainage of the postglacial North Sea terrestrial**
**area. LWNSL = Late Weichselian North Sea Lake. NSPP = North Sea Palaeolandscapes Project (Gaffney et al., 2007, 2009).**