# Peer review of "Ice sheet and palaeoclimate controls on drainage network evolution: an example from Dogger Bank, North Sea"

_Earth Surface Dynamics, 2020_

## Referee Comment (RC1) · Anonymous Referee #1 · 7 Jul 2020

General comments

Very interesting manuscript that presents new results from subsurface mapping at the Dogger Bank. The new results are used soundly by the authors to discuss implications for the geological development in the central North Sea region since the LGM, and for drainage system evolution in general when subjected to environmental changes from glacial to marine. The quality of the data used is generally very high with a dense grid of high-resolution reflection seismic data and many CPTs.

However, some of the figures could be improved to increase clarity and documentation (ex by use of a, b, c, for 'subparts' of figures and with reference to such 'subparts' in

the text) (see more in specific comments) and there are a few issues that could be elaborated (or alternatively omitted):

1) Various links to subglacial valleys discussed or highlighted in the manuscript. These links often appear somewhat contradictory. a. Channel 1 and 2 resemble subglacial valleys in their dimensions and undulating base and the authors show a deeper subglacial valley in Fig. 9 that has a very similar appearance as channel 1 b. On the other hand, the authors argues that there is a lack of subglacial valleys in the study area c. The course of channel 1 is explained by the underlying subglacial valley but it is not further accommodated in the presented model of formation 2) The erosional features at the seabed is not well documented or discussed. Maybe they are the subject of another manuscript in preparation. It would however be good to further document them - e.g. by adding a cross-section to Fig. 9 showing both the subglacial valley and the erosional features.

Listed below are some further specific comments to the text. Despite the many comments, the manuscript is generally of very high quality. It is very well written, with a logic structure and a clear focus, and it represents a great contribution to geological research of the Dogger Bank.

Specific comments Line 65- 81: How certain are the ages provided? 23 kyr BP, 8 kyr BP Line 80-81: "..buried during subsequent marine transgression around 10 ka". This contrasts Fig. 8 where marine transgression is stated to happen at ca. 9 ka BP Line 95: Is it multichannel or single channel reflection seismic data? Line 98: Was a bandpass filter the only processing carried out? Seems a bit too little to do on data if they are multichannel sparker data. If single channel data - ok. Please clarify. Bandpass filtering can also be done in IHS Kingdom Suite Line 100: state velocity used for calculation of vertical resolution in meters. Line 103: Please clarify how you derived the velocity from the geotechnical data (CPTs) – or refer to Cotterill et al 2017a here also. Line 105: the grid cell size of 10 m x 10 m appears to be a bit small (at least for the 'across' line direction) considering that your line spacing is

at best 100 m. Please elaborate on the choice of cell size. Maybe by including the horizontal resolution of the seismic data (inline direction)? Line 106: which further processing in QGIS is referred to here? Specify. Line 113-116: you have used cone resistance as a proxy for grain size. Did you calculate soil behavior type index? Or plotted cone resistance to friction ratio (classic Robertson diagram)? Did you normalize and correct the measured values for pore pressure and burial depth? Please specify. Line 118: Why not just do this in Kingdom Suite? Line 119-120: "Using this method the upstream. . ." What do you mean by this sentence? Elaborate Line 121: UTM31N - consider to also state the geodetic datum. WGS 84? Line 125: Please consider the uncertainty related to the gridding at 10 m cells here. In principle only every 10th point would be data based if the channel direction was perpendicular to your inlines (with line spacing of 100 m). Line 127-143: The two models are significantly different from 26 to 18 ka BP. It is hard to provide better modelling, as the authors stress, but the uncertainty for this period and the potential for other environmental interpretations should probably be considered in more detail. Line 149: Please highlight horizon Z in the lower panel of Fig. 1. Line 151- 157: Refer to Fig. 2 Line 153 and elsewhere: Consider to use 'reflections' and not 'reflectors' for what you see on the seismic cross-sections. Reflectors are the physical boundaries in the subsurface while reflections are the geophysical representations of these reflectors. Line 165: reference to one of your figures (e.g. Fig. 4b?) Line 235+316+368: very hard to see these low-relief valleys from the maps you have presented. Do you just mean the centerline of the main channel segments or do you mean wider valleys in Horizon Z? Specify or show better on figures (i.e. use another color scale for the grid of Horizon Z). Line 237-238: consider to refer to Prins & Andresen (2019) that discuss a subglacial valley origin of their river channels. Line 238-239: consider to refer to the Ottesen et al 2020 on tunnel valleys in the North Sea (https://doi.org/10.1016/j.margeo.2020.106199) Line 236-243: In this section you present your interpretation of the channels as fluvial rivers. The dimensions and stratigraphic position of the channels and the lack of deformation of underlying sediments are presented as arguments for the fluvial origin. However, the

similarity in terms of dimensions, between channel 1 and the underlying subglacial valley (Fig. 9) is not accounted for here. Could this similarity support a later ice-sheet readvance for initiation of channel 1 and 2? Elaborate. Line 264: for consistency, use 'channel-fills' instead of 'channels' here. Line 271-272: How can you see that the dipping reflections are downstream? It appears that the cross-sections you show are mainly across (perpendicular to) the channels and not along. Maybe you have other lines showing the downstream direction better? Please refer to a specific figure here (e.g. Fig. 4c) Line 275: 'streams' not used previously. Maybe better just to stick to the 'tributary' term Line 280-281: deepest point. Depth measures relative to mean sea level or? Consider to add a remark on the uncertainty in the grids and the undulating basal profiles Line 282-283: Not always easy to see from Fig. 7, that the tributaries steepen towards the confluence of the main channels. Line 314-315: Consider to add a cross-section to Fig. 9 where you show the deeper subglacial valley and the erosional features at the seafloor. This would make your argumentation stronger Line 316-317: How does channel 1 fit into this argument? Line 345: please specify which CPTs in Figure 3 Line 358-363: the lack of tunnel valleys does not fit with your own observation of a subglacial valley (Fig. 9). Please include in your argumentation. Line 374: please highlight these flat areas on your figures (ex fig. 4 or 6?) Line 377: consider to use another wording than 'best' Line 395-396: Show these erosional features on a cross section – maybe as add-on to Fig. 9. Would be interesting to see how they look in order to assess the interpretation as tidal scours. Line 415: consider to add Bølling-Allerød in brackets after 15 ka BP. Line 416-439: The very large differences in the two model runs from 26-18 ka BP are partly discussed and accounted for in the text. However, it would be good to further describe the uncertainties in the two models – and in turn the uncertainties for the presented environmental interpretations. How would a more humid environment from 26-18 ka BP fit/change your model of formation. Line 450-455: Description does not fully match what is shown in Fig. 11. Line 450: westward instead of eastward?

Fig. 1: Subdivide into Fig. 1a (map) and Fig. 1b (seismic cross-section and interpretation). Highlight Horizon Z and sub-unit 1,2,3 in b). Comment on the incision in the seabed. Is this the erosional feature discussed later or? Fig. 2: Subdivide into Fig 2a (facies map) and Fig. 2b (Isopach upper seismic unit). Please add location of seismic line shown in Fig. 1b on both Fig. 2a and 2b. Add outline box for location of Fig. 9. Caption: A bit confusing with the formulation "subcrop map of the major basal seismic unit facies" Consider to rephrase to "seismic facies of the basal seismic unit subcropping Horizon Z" Fig. 3: Very busy figure. Subdivide into Fig 3a (map) Fig. 3b (seismic cross-section) and Fig 3c: (CPTs). Map (Fig 3a): Why data gap in map? - Briefly explain in figure caption. Add outline boxes for Fig 4 and 5 maps. Hard to read CPT names – consider to use white box background. Hard to see location of section A-A' CPTs (Fig. 3c): Consider to reorder the shown CPTs to a more logical arrangement. Ex sorted by penetration of main channels or tributaries, or no channels. Please place the name label for the CPTs consistently in one area of the logs. Fig. 4: Subdivide into Fig 4a (map), Fig 4b (cross-section A-A') Fig 4c (Cross-section B-B') Map: add location of Fig 3b cross-section. Add CPTs. Explain white dashed lines Fig. 5: Subdivide in to Fig 5a (map), Fig 5b (cross-section A-A') Fig 5c (Cross-section B-B') Map: add CPTs Fig 5b+c. hard to see whether it is the dark or pale green color shown. Highlight in caption. Fig. 6: Add outline box for map shown in Fig. 9. Where is IC3? What about unnamed ICs (north of Channel 1) Fig. 7: Use subdivision a, b, c, d, e. a, b: Concerning the number of channels analyzed a bit more explanation would be good. The six isolated channels probably is IC1,2,4,5,6,7 which is ok but should be specified for the reader in the caption, The number of tributaries is 8 but I can only count 7 on the map in Fig. 6. Please clarify. c, d, e: Please clarify what OD is (elevation (m OD)). Hard to see steepening into main channels for all of the tributaries Fig, 8: Green colors are here used for proglacial lake fill. A bit confusing when comparing to the cross-sections where green colors are used to indicate channel fill. Consider to change or show a legend for the colors. Fig. 9: Please comment on the fact that this tunnel valley has similar dimensions as channel 1 – meaning that dimensions may not be a valid argument for a fluvial origin of your channels. The 'lack of subglacial valleys in the area'

argumentation should take into account your own subglacial valley. Please include a cross section to show the tunnel valley and the erosional features Fig. 11: Subdivide into 11a (Map) and 11b (timing of events). Map: directions of drainage outlets do no match what is stated in the text. Ex: Paleo-Ems (Hepp et al 2019) goes into EPV. Add channels from Prins & Andresen (2019) instead of just study area.

Technical corrections Line 4: Include F. in author name Ruza Ivanovic to match what is stated in acknowledgement line 504 Line 119. New sentence after 'truncated'. So "...forms where underlying reflections are truncated. Using this method..." Line 502: Include these shape-files of the channels from Prins & Andresen (2019) in Fig. 11a

---

## Referee Comment (RC2) · Anonymous Referee #2 · 10 Jul 2020

General comments

In the manuscript Emery and coauthors study the postglacial paleolandscape evolution of Dogger Bank by using geophysical and CPT data and try to understand the involved paleoclimate controls by utilizing paleoclimatic modeling. As they study the changes of the Earth's surface and the influencing factors, the manuscript is well within scope of the journal Earth Surface Dynamics.

The study is well designed and based on a wealth of geophysical and geological data. The results are presented in a clear and concise manner and their interpretation is well argued in later sections of the manuscript. To summarize, the authors pre-

pared a very interesting paper which will allow the community to better understand the postglacial environments and evolution in the North Sea. The paper undoubtedly represents a valuable contribution in the understanding of this region.

My only major concern regarding this manuscript is the lack of any chronostratigraphical data. However, as regional chronostratigraphical constraints are well established and the authors take them in account, this is not a critical limitation of the paper. There are two other minor issues that are also described in the Specific remarks. Firstly, in parts of the paper (especially in the sections describing the results of the seismic interpretation) the authors very vaguely present where a described feature is shown in figures. For example, they refer only to a figure number. These often contain 2 geophysical profiles which leaves the reader struggling with finding out to which profile the authors were referring. I suggest the authors modify the manuscript in order to make the reading a bit easier. Secondly, I have some suggestions regarding the artwork. Generally it looks very nice; I appreciate the use of uninterpreted and interpreted profiles in the same figure and I really approve of the use of "scientific colours". However, the reader would really benefit, if locations of maps from Figs. 4 and 5 would be included in a study area figure (in Fig. 3 or 6, for example). In addition, the figures are sometimes a bit cramped (see Specific remarks for Fig. 1).

Overall, in my opinion the authors prepared a very good and interesting paper which only needs some minor modifications before publication.

Specific remarks

L113: Maybe a reference to Fig. 3 would be suitable in this part of the manuscript to refer the reader to the CPT locations?

L136-142: "The GLAC-1D ice-sheet . . . representation of climate thereafter." – this part of the manuscript could be a part of the discussion section.

Section 4: The results of palaeoclimate modelling are not presented in the Results

section. I suggest the authors dedicate a sentence or two to these results (maybe refer the reader to Fig. 10).

Section 4.1.1: The authors could refer to Fig. 2A in this part of the manuscript.

Section 4.1.2: The authors could provide the figures depicting the different appearances of Horizon Z (e.g. "... coincident with the seabed (profile A-A' in Fig. 5)."

Section 4.1.3: Similarly to the previous comment, the authors could provide the figures depicting the different types of appearances of the channel fill on the seismic sections (for example, for the acoustic blanking). Additionally, it would be beneficial for the readers, if the authors specify more in detail, where different details of the acoustic facies can be observed. The description between L171-172 could be "(middle part between 38 and 28 ms on profile A-A' in Figure 4)" instead of just "Figure 4". The described details are sometimes difficult to find in the figures (for example, I don't see the prograding fill in Fig. 4 and I don't even know which profile to observe).

Section 4.1.4: Again, I suggest the authors provide more in detail where the described features of the acoustic facies can be observed in the profiles.

L189-190: "In the north of the study area, the largest elongate feature can be observed to incise through the channel-fill unit and into the basal seismic unit" I suggest the authors refer the reader to a figure with a map which demonstrates this.

L293: Subdued by what process? I suggest the authors use a word that is more descriptive or also reflects the possible process (e.g. relief was eroded, compacted, leveled out...).

L305: A reference to a figure would be appropriate after "accretion".

L307: Maybe "1st part of Figure 8"

L309: Do the authors have any idea, why the widths are relatively narrow and constant? Is it possible, that they were previously confined by relief (which is not preserved?). Are

there any indices in the geophysical datasets for this?

L316: The authors want to demonstrate that Channel 2 was located in a valley and probably mistakenly refer to the isopach map (Fig. 2B) instead to the horizon-depth map (map in Fig. 3). Nevertheless, I am not really convinced from Fig. 3 that channel 2 is located in a valley as it seems to be located on a topographical high of Horizon Z (between -30 and -33 m).

L343: On which profile and where specifically is cross-bedding visible in Fig. 4?

L377-378: The authors state that the warmer-climate drainage network is best developed over the proglacial lake-fill sediments, however, the largest feature (Channel 3) is developed outside the bounds of the proglacial lake.

L396-397: "sigmoidal to oblique reflectors in the upper seismic unit" - a reference to a figure would be appropriate in this part of the manuscript. "infill of the channels and the tidal scour features" – a figure showing a profile across the proposed tidal scours would be appropriate in this part of the manuscript.

L487-489: "Palaeoclimate modelling showed a cold, arid period between ice sheet retreat at 23 ka BP and 17 ka BP, when the climate became increasingly warm and wet, which correlates to marsh environments at Dogger Bank c. 14.9 – 13.5 ka BP." – Correlates in what way? A part of the sentence seems to be missing, as a cold period is regarded as a warm period. Or should "when" be "then"?; maybe "during ice sheet retreat between" instead of "between ice sheet retreat at"

L772-773: As this report is cited very often and is available online, I suggest the authors add the hyperlink to the report, if the journal guidelines allow.

Fig. 1: The font for the scale bar is disproportionally large compared to the other text on the figure. I also suggest to put the text for the depth and elevation colourbars below the colorbars. In that way, both texts are physically separated from the Forewind and Study area part of the legend and the legend becomes clearer. But these are just my

personal preferences...

Maps in Fig. 4, 5 and 9: It would be really beneficial for the readers to have the location of Figs. 4, 5 and 9 marked on one of the smaller scale Figures (Fig. 2B or Fig. 6 or . . .).

Fig. 3: What is the "m OD" abbreviation on the Horizon Z depth map?

L786: Personally, I really appreciate the authors using "scientific colours" and hope others will follow.

Fig. 9: In L100 the authors mention that the reflections can be recognised up to 150 m deep. However, according to Fig. 9, the tunnel valley is more than 200 meters deep. If this is a mistake, the authors should correct the figure, otherwise I suggest you also include a reference to a previous study or a figure with a profile showing the tunnel valley (possibly 2 profiles to show the relation of the uneroded and eroded channel with the valley).

Fig. 11: The location of Oyster Ground is not marked on the map

Technical corrections

L18: probably "represent a terrestrial" instead of "represent terrestrial"?

L19: "comprises" instead of "comprise"

L28: probably "9 ka BP" instead of "8 ka BP"

L108: maybe "and the extended interpretation of" instead of "and extended for interpretation of"

L114: "proxy for" instead of "proxy to"; alternatively you could use grain-size proxy

L119: "truncated. Using" instead of "truncated, using"

L151: probably "Generally the area" instead of "Generally the"?

L160-161: "Figure 2" should be "Figure 2b"?

L186: Is "Figure 2" appropriate here? As "(Figure 2)" is placed at the end of the sentence, it seems that the authors are referring to a seismic profile and not an isopach map. If they are indeed referring to the map, I suggest they put "(Figure 2B)" after "thickest" in L185.

---

## Author Comment (AC1) · 26 Aug 2020

General comments

Very interesting manuscript that presents new results from subsurface mapping at the Dogger Bank. The new results are used soundly by the authors to discuss implications for the geological development in the central North Sea region since the LGM, and for drainage system evolution in general when subjected to environmental changes from glacial to marine. The quality of the data used is generally very high with a dense grid of high-resolution reflection seismic data and many CPTs. However, some of the figures could be improved to increase clarity and documentation (ex by use of a, b, c, for 'subparts' of figures and with reference to such 'subparts' in the text) (see more in specific comments) and there are a few issues that could be elaborated (or alternatively omitted):

1) Various links to subglacial valleys discussed or highlighted in the manuscript. These links often appear somewhat contradictory. a. Channel 1 and 2 resemble subglacial valleys in their dimensions and undulating base and the authors show a deeper sub glacial valley in Fig. 9 that has a very similar appearance as channel 1 b. On the other hand, the authors argues that there is a lack of subglacial valleys in the study area c. The course of channel 1 is explained by the underlying subglacial valley but it is not further accommodated in the presented model of formation 2) The erosional features at the seabed is not well documented or discussed. Maybe they are the subject of another manuscript in preparation. It would however be good to further document them - e.g. by adding a cross-section to Fig. 9 showing both the subglacial valley and the erosional features.

Listed below are some further specific comments to the text. Despite the many comments, the manuscript is generally of very high quality. It is very well written, with a logic structure and a clear focus, and it represents a great contribution to geological research of the Dogger Bank.

We thank the reviewer for their detailed, constructive, and useful comments. We have addressed all of the specific comments below. Please note the line numbers below refer to the tracked changes pdf version of the revised manuscript.

As requested, we have added parts A, B, C, etc. to figures to make it clearer to which part the manuscript is referring to. We have clarified our justification for a fluvial origin for the channels, and tidied up contradictory text to make it clear which channels we are referring to and when. The erosional features are not discussed at great length in this paper as they will be discussed in another paper that covers the coastal to fully marine transition, but a cross-section has been added (Figure 9) to illustrate their relationship to the river channels.

Specific comments

Line 65- 81: How certain are the ages provided? 23 kyr BP, 8 kyr BP

These ages are taken from published literature, and absolute uncertainties within the given timing are presented within those studies. However, we have added wording to reflect the fact that the dates are relatively uncertain (e.g. line 77, lines 84-85)

Line 80-81: "..buried during subsequent marine transgression around 10 ka". This contrasts Fig. 8 where marine transgression is stated to happen at ca. 9 ka BP

We have reworded this sentence to clarify meaning, line 84-85: "The channel network was buried during subsequent marine transgression, which began at around 10 ka BP at Dogger Bank, with complete inundation occurring around 8.5-8 ka BP"

Line 95: Is it multichannel or single channel reflection seismic data?

Single-channel. We have added this clarification (line 104)

Line 98: Was a bandpass filter the only processing carried out? Seems a bit too little to do on data if they are multichannel sparker data. If single channel data - ok. Please clarify. Bandpass filtering can also be done in IHS Kingdom Suite

We have expanded this sentence to include the full processing flow, lines 107-108: "Data were recorded in StrataView, then imported to ProMAX for processing, where a bandpass filter with a 100 Hz lowcut and an 800 Hz highcut was applied, followed by F-k filtering and time migration, then exported to SEG-Y"

Line 100: state velocity used for calculation of vertical resolution in meters.

We have added the velocity of 1600 m/s (line 110)

Line 103: Please clarify how you derived the velocity from the geotechnical data (CPTs) – or refer to Cotterill et al 2017a here also.

These data are described in Cotterill et al. 2017a, so a reference has been added (line 114)

Line 105: the grid cell size of 10 m x 10 m appears to be a bit small (at least for the 'across' line direction) considering that your line spacing is at best 100 m. Please elaborate on the choice of cell size. Maybe by including the horizontal resolution of the seismic data (inline direction)?

We have added a justification for using 10 m cell size, along with details on lines 116-118: "A 10 m grid was chosen as a compromise to maintain the necessary level of detail from the high horizontal resolution (0.73 m at 1600 m/s) along the seismic lines, whilst maintaining a reasonable correlation distance between data points on lines spaced at 100 m."

Line 106: which further processing in QGIS is referred to here? Specify.

We replaced the words "further processing" with "interpretation and display" to clarify what was done in QGIS (line 116).

Line 113-116: you have used cone resistance as a proxy for grain size. Did you calculate soil behavior type index? Or plotted cone resistance to friction ratio (classic Robertson diagram)? Did you normalize and correct the measured values for pore pressure and burial depth? Please specify.

No corrections were made. We simply used the cone resistance (qc) values as a grain-size proxy. Although this does not give detailed CPT-stratigraphic information, it gives sufficient stratigraphic variation to correlate to, and differentiate between, different seismic units. We have reworded the sentence to clarify this method, also used in Emery et al., 2019a, on lines 126-128: "These tests provide cone resistance (qc) measurements that were used, uncorrected, as a grain-size proxy through the sediments, with low resistance corresponding to clay and high resistance corresponding to sand, as used by Emery et al. (2019a)."

Line 118: Why not just do this in Kingdom Suite?

We prefer the use of QGIS for digitising shapefiles, as it allows for more flexibility and tools such as merging and joining individual shapefiles, and geometry analysis, and also is more flexible in creating final maps in QGIS.

Line 119-120: "Using this method the upstream..." What do you mean by this sentence? Elaborate

We removed this sentence as it was a mistakenly left in from an earlier draft.

Line 121: UTM31N - consider to also state the geodetic datum. WGS 84?

Yes, WGS84. This information has been added (line 142).

Line 125: Please consider the uncertainty related to the gridding at 10 m cells here. In principle only every 10th point would be data based if the channel direction was perpendicular to your inlines (with line spacing of 100 m).

We added the fact that these profiles were visually smoothed to remove any such interpolation biases on lines 145-146: "As the seismic horizon was gridded at 10 m, long profile points were automatically extracted by the QGIS Profile Tool plugin every 10 m, then visually smoothed to remove effects of seismic line mistie and any interpolation bias."

Line 127-143: The two models are significantly different from 26 to 18 ka BP. It is hard to provide better modelling, as the authors stress, but the uncertainty for this period and the potential for other environmental interpretations should probably be considered in more detail.

We have considered this in relation to your later point about lines 416-439, and discuss it there.

Line 149: Please highlight horizon Z in the lower panel of Fig. 1.

We have added Horizon Z to Figure 1.

Line 151- 157: Refer to Fig. 2

We have added a reference to Figure 2 (lines 182-188).

Line 153 and elsewhere: Consider to use 'reflections' and not 'reflectors' for what you see on the seismic crosssections. Reflectors are the physical boundaries in the subsurface while reflections are the geophysical representations of these reflectors.

We have replaced the word "reflectors" with "reflections" throughout the manuscript to improve the use of correct terminology.

Line 165: reference to one of your figures (e.g. Fig. 4b?)

We have added a reference to Figure 3B (line 196).

Line 235+316+368: very hard to see these low-relief valleys from the maps you have presented. Do you just mean the centerline of the main channel segments or do you mean wider valleys in Horizon Z? Specify or show better on figures (i.e. use another color scale for the grid of Horizon Z).

We have added a new part to Figure 3, using cross-section profiles to show the low-relief valleys.

Line 237-238: consider to refer to Prins & Andresen (2019) that discuss a subglacial valley origin of their river channels.

We have extended the sentence and referenced Prins & Andresen (2019) here to explain the potential origin of the river channels as subglacial (lines 316-317): "The channels might have

originated as a tunnel valley network, such as that interpreted to the east of Dogger Bank (Prins and Andresen, 2019)."

Line 238-239: consider to refer to the Ottesen et al 2020 on tunnel valleys in the North Sea ()

We have added this reference, and a reference to Prins et al. (2020), which describe North Sea tunnel valleys in more detail (line 329).

Line 236-243: In this section you present your interpretation of the channels as fluvial rivers. The dimensions and stratigraphic position of the channels and the lack of deformation of underlying sediments are presented as arguments for the fluvial origin. However, the similarity in terms of dimensions, between channel 1 and the underlying subglacial valley (Fig. 9) is not accounted for here. Could this similarity support a later ice-sheet readvance for initiation of channel 1 and 2? Elaborate.

The tunnel valley shown on Figure 9 is a similar width to channels 1 & 2, but it is up to 150 m deep in places, unlike the interpreted river channels, which reach a maximum depth of 15 m. An extra part to show the difference in morphology between the two channels, also showing the different stratigraphic levels of the two channel types, has been added to Figure 9. Whilst a subglacial origin for the channels cannot be ruled out, we feel the lack of evidence for glaciation at the stratigraphic level of the river channels, located above proglacial sediments, is sufficient evidence that they did not form subglacially. We have expanded our discussion of the stratigraphy, sedimentology and geomorphology to back up our choice of fluvial origin (lines 331-335): "However, no evidence of deformation related to readvance is recorded within the proglacial lake-fill sediments or anywhere else throughout Tranche B, and there are no glaciogenic deposits or glacial geomorphology at the stratigraphic level of the channel network (Emery et al., 2019a). Subglacial channels in the Dogger Bank area are either smaller (10s of m wide; Emery et al., 2019a) or of a similar width to these channels, but markedly deeper, up to 100 m deep (Figure 9). Therefore, we favour a fluvial origin for these channels that incised into Horizon Z."

Line 264: for consistency, use 'channel-fills' instead of 'channels' here.

We have changed this to "channel-fills" (line 356).

Line 271-272: How can you see that the dipping reflections are downstream? It appears that the cross-sections you show are mainly across (perpendicular to) the channels and not along. Maybe you have other lines showing the downstream direction better? Please refer to a specific figure here (e.g. Fig. 4c)

We have added a reference to Figure 4C here (line 368), which shows dipping downstream obliquely through the channel. We have also added annotation to Figure 4C to make this clearer. There are no crosslines that intersect parallel to the channel to show downstream-dipping reflections available in the dataset.

Line 275: 'streams' not used previously. Maybe better just to stick to the 'tributary' term

We have deleted "and streams" to avoid confusion (line 371).

Line 280-281: deepest point. Depth measures relative to mean sea level or? Consider to add a remark on the uncertainty in the grids and the undulating basal profiles

Depth relative to the edge of the channels. We have extended the sentence to include the smoothing procedure that reduces the uncertainty in the grids, so that the undulating basal profiles

are real (lines 376-378): "Long profiles of the three main channels and their longest tributaries were drawn from the centre-lines of the deepest point of the channel base relative to the channel edge, and smoothed to reduce issues of seismic mistie and interpolation bias (Figure 7)."

Line 282-283: Not always easy to see from Fig. 7, that the tributaries steepen towards the confluence of the main channels.

We have added the caveat that only some tributaries steepen (lines 379-381), and added annotations to Figure 7: "The tributary channel bases also decrease in elevation (Figure 7), and sometimes become steeper from the tributary head to the confluence with main channels, such as in main channel 1 (Figure 7), implying these channels were cutting down to the main channels."

Line 314-315: Consider to add a cross-section to Fig. 9 where you show the deeper subglacial valley and the erosional features at the seafloor. This would make your argumentation stronger

We have added a cross-section of the relationship between the tunnel valley and river channels, and the relationship between the erosional features and river channels, to Figure 9.

Line 316-317: How does channel 1 fit into this argument?

This is a separate topographic control, and as such we have reworded the sentence to make this clearer (lines 392-393): "Antecedent topography also affected the location of Channel 2, which flowed down the axis of the former proglacial lake, and is located at the base of a shallow valley (Figure 3D)."

Line 345: please specify which CPTs in Figure 3

We have added the CPTs to this figure reference (line 481): "(e.g. CPTs H, K, O, and W; Figure 3)"

Line 358-363: the lack of tunnel valleys does not fit with your own observation of a subglacial valley (Fig. 9). Please include in your argumentation.

We have added a sentence to explain the chronostratigraphic separation between this tunnel valley and the river channels, and that the tunnel valley was potentially formed much earlier than the river channels (lines 506-508): "The tunnel valley adjacent to, and controlling the location of, Channel 1 (Figure 9) is of unknown age, but predates stratigraphy related to MIS 2 ice-sheet retreat, and may be related to ice-sheet advance during MIS 3."

Line 374: please highlight these flat areas on your figures (ex fig. 4 or 6?)

A reference to Figure 3B has been added (line 519) and highlighted on Figure 3B.

Line 377: consider to use another wording than 'best'

We replaced the word "best" with "most" (line 523)

Line 395-396: Show these erosional features on a cross section – maybe as add-on to Fig. 9. Would be interesting to see how they look in order to assess the interpretation as tidal scours.

These erosional features have been added to Figure 9.

Line 415: consider to add Bølling-Allerød in brackets after 15 ka BP.

We added "(Bølling-Allerød Interstadial)" to the end of the sentence (line 579).

Line 416-439: The very large differences in the two model runs from 26-18 ka BP are partly discussed and accounted for in the text. However, it would be good to further describe the uncertainties in the two models – and in turn the uncertainties for the presented environmental interpretations. How would a more humid environment from 26-18 ka BP fit/change your model of formation.

We have further explored the differences in the model runs, and considered implications of different palaeoclimates, in lines 616-621. This considers uncertainties and the balances between desiccation and drainage network formation, and clarifies our justification for a change of climate at around 17 ka BP: "The uncertainty in the early part of the palaeoclimate simulations due to ice-sheet models, from 26 – 17 ka BP, could potentially allow for a change to a more humid, higher-precipitation climate from earlier than 17 ka BP. This would allow more time for the dendritic drainage channel network to form, but less time with little precipitation to allow for the desiccation of the glaciogenic sediments. On the balance of these two factors, we prefer the interpretation of 17 ka BP being the onset of a warmer climate with more precipitation, that initiated marshland and the drainage network on top of desiccated proglacial lake-fill and glacial outwash sediments."

Line 450455: Description does not fully match what is shown in Fig. 11.

We have added some annotation to Figure 11 to clarify the link between what is described in the text and what is shown on the figure.

Line 450: westward instead of eastward?

Yes, amended (line 632)

Fig. 1: Subdivide into Fig. 1a (map) and Fig. 1b (seismic cross-section and interpretation). Highlight Horizon Z and sub-unit 1,2,3 in b). Comment on the incision in the seabed. Is this the erosional feature discussed later or?

We have subdivided Figure 1 into A and B. We have highlighted horizon Z, and annotated basal sub-units 1, 2 and 3 onto the section. We have also annotated the difference between present-day seabed scour and the erosional features discussed in the text.

Fig. 2: Subdivide into Fig 2a (facies map) and Fig. 2b (Isopach upper seismic unit). Please add location of seismic line shown in Fig. 1b on both Fig. 2a and 2b. Add outline box for location of Fig. 9. Caption: A bit confusing with the formulation "subcrop map of the major basal seismic unit facies" Consider to rephrase to "seismic facies of the basal seismic unit subcropping Horizon Z"

We have subdivided Figure 2 and added the seismic line location. We added the location of Figure 9 to Figure 3. We have rephrased the caption for Figure 2A as suggested.

Fig. 3: Very busy figure. Subdivide into Fig 3a (map) Fig 3b (seismic cross-section) and Fig 3c: (CPTs). Map (Fig 3a): Why data gap in map? - Briefly explain in figure caption. Add outline boxes for Fig 4 and 5 maps. Hard to read CPT names – consider to use white box background. Hard to see location of section A-A' CPTs (Fig. 3c): Consider to reorder the shown CPTs to a more logical arrangement. Ex sorted by penetration of main channels or tributaries, or no channels. Please place the name label for the CPTs consistently in one area of the logs.

We have split Figure 3 into parts A, B, and C, and added a part D. 3A: Some seismic lines are missing from the dataset. We have added the locations of Figures 4,5 and 9 to the map. The outline around CPT names on the map has been thickened to increase legibility, and labels rearranged to show A-A' more clearly. 3C: We have left the CPTs in the same order as they run clockwise around the map. We

have added this ordering into the caption. It is not possible to move the name label into one consistent place across the logs as that obscures data.

Fig. 4: Subdivide into Fig 4a (map), Fig 4b (cross-section A-A') Fig 4c (Cross-section B-B') Map: add location of Fig 3b cross-section. Add CPTs. Explain white dashed lines

We have subdivided Figure 4, added the Figure 3B location and CPTs, and annotated what the white dashed lines are (channel thalwegs).

Fig. 5: Subdivide in to Fig 5a (map), Fig 5b (cross-section A-A') Fig 5c (Cross-section B-B') Map: add CPTs Fig 5b+c. hard to see whether it is the dark or pale green color shown. Highlight in caption.

We have subdivided Figure 5. We have added CPTs to the map. We have altered the key to show the coarse channel-fill.

Fig. 6: Add outline box for map shown in Fig. 9. Where is IC3? What about unnamed ICs (north of Channel 1)

The outline of Figure 9 is now shown on Figure 3. We have relabelled the ICs to include 3 as it was mistakenly missed. The unnamed ICs were not included in the analysis as they are too short in the dataset.

Fig. 7: Use subdivision a, b, c, d, e. a, b: Concerning the number of channels analyzed a bit more explanation would be good. The six isolated channels probably is IC1,2,4,5,6,7 which is ok but should be specified for the reader in the caption, The number of tributaries is 8 but I can only count 7 on the map in Fig. 6. Please clarify. c, d, e: Please clarify what OD is (elevation (m OD)). Hard to see steepening into main channels for all of the tributaries

We have subdivided Figure 7. We have recounted the number of ICs and tributaries and it is 7 and 7, not 8 and 6 as previously mistakenly stated. We have removed OD from the figures as it is confusing for international readers. We have annotated the steepening into the channels where present.

Fig, 8: Green colors are here used for proglacial lake fill. A bit confusing when comparing to the cross-sections where green colors are used to indicate channel fill. Consider to change or show a legend for the colors.

We have changed the green proglacial lake fill to a purple.

Fig. 9: Please comment on the fact that this tunnel valley has similar dimensions as channel 1 – meaning that dimensions may not be a valid argument for a fluvial origin of your channels. The 'lack of subglacial valleys in the area' argumentation should take into account your own subglacial valley. Please include a cross section to show the tunnel valley and the erosional features

We have added a cross-section of the tunnel valley to show how it differs to the river channels (as previously discussed in this response).

Fig. 11: Subdivide into 11a (Map) and 11b (timing of events). Map: directions of drainage outlets do no match what is stated in the text. Ex: Paleo-Ems (Hepp et al 2019) goes into EPV. Add channels from Prins & Andresen (2019) instead of just study area.

We have annotated this figure to match what is stated in the text, as the drainage pattern evolves over time. We have added the shapefile from Prins & Andresen (2019).

Technical corrections

Line 4: Include F. in author name Ruza Ivanovic to match what is stated in acknowledgement line 504

We have added the F as requested (line 4).

Line 119. New sentence after 'truncated'. So "...forms where underlying reflections are truncated. Using this method..."

As previously discussed, we have removed the send part of this sentence as it was mistakenly left in from an earlier draft.

Line 502: Include these shape-files of the channels from Prins & Andresen (2019) in Fig. 11a

We have added the shapefile from Prins & Andresen (2019) to Figure 11.

---

## Author Comment (AC2) · 26 Aug 2020

General comments

In the manuscript Emery and coauthors study the postglacial paleolandscape evolution of Dogger Bank by using geophysical and CPT data and try to understand the involved paleoclimate controls by utilizing paleoclimatic modeling. As they study the changes of the Earth's surface and the influencing factors, the manuscript is well within scope of the journal Earth Surface Dynamics.

The study is well designed and based on a wealth of geophysical and geological data. The results are presented in a clear and concise manner and their interpretation is well argued in later sections of the manuscript. To summarize, the authors prepared a very interesting paper which will allow the community to better understand the postglacial environments and evolution in the North Sea. The paper undoubtedly represents a valuable contribution in the understanding of this region.

My only major concern regarding this manuscript is the lack of any chronostratigraphical data. However, as regional chronostratigraphical constraints are well established and the authors take them in account, this is not a critical limitation of the paper. There are two other minor issues that are also described in the Specific remarks. Firstly, in parts of the paper (especially in the sections describing the results of the seismic interpretation) the authors very vaguely present where a described feature is shown in figures. For example, they refer only to a figure number. These often contain 2 geophysical profiles which leaves the reader struggling with finding out to which profile the authors were referring. I suggest the authors modify the manuscript in order to make the reading a bit easier. Secondly, I have some suggestions regarding the artwork. Generally it looks very nice; I appreciate the use of uninterpreted and interpreted profiles in the same figure and I really approve of the use of "scientific colours". However, the reader would really benefit, if locations of maps from Figs. 4 and 5 would be included in a study area figure (in Fig. 3 or 6, for example). In addition, the figures are sometimes a bit cramped (see Specific remarks for Fig. 1).

Overall, in my opinion the authors prepared a very good and interesting paper which only needs some minor modifications before publication.

We thank the reviewer for their clear, insightful, and constructive review comments. We have replied individually to specific remarks and technical corrections below.

We agree with the reviewer that a lack of chronostratigraphic control is frustrating when dealing with landscape evolution in offshore areas, and greater chronostratigraphic constraint would be beneficial. However, given the regional chronostratigraphic constraints and the high-resolution physical stratigraphy, this does not cause too much of an issue for the scope of this manuscript. We are confident that future workers will be able build on our work, employing tighter chronostratigraphic constraints.

Throughout the manuscript, we have improved both the referencing of specific parts of figures using letters to denote parts, and improved annotations on the figures themselves to draw the reader's attention to the necessary feature.

We have cross-referenced the locations of figure 4, 5, and 9 on figure 3, and attempted to make figures less cramped by rearranging some keys and annotations. Please note the line numbers below refer to the tracked changes pdf version of the revised manuscript.

Specific remarks

L113: Maybe a reference to Fig. 3 would be suitable in this part of the manuscript to refer the reader to the CPT locations?

We have added a reference to Figure 3 (line 126)

L136-142: "The GLAC-1D ice-sheet ... representation of climate thereafter." – this part of the manuscript could be a part of the discussion section.

We agree, and we have moved this section and integrated it into the discussion (lines 580-586)

Section 4: The results of palaeoclimate modelling are not presented in the Results section. I suggest the authors dedicate a sentence or two to these results (maybe refer the reader to Fig. 10).

We have added a section 4.5 to briefly describe results of the palaeoclimate modelling (lines 386-393): "4.5 Palaeoclimate modelling

The palaeoclimate simulation outputs for the two model runs using GLAC-1D and ICE-6G_C ice sheet reconstructions for the timespan of 26 ka BP to present are shown in Figure 10. Generally, the climate simulations show similar trends through the Holocene, but differ through the Late Pleistocene. The climate simulation using GLAC-1D has much higher precipitation than the equivalent simulation with ICE-6G_C between 26 and 18 ka BP, but the climate with ICE-6G_C shows much higher precipitation than with GLAC-1D between 18 and 11 ka BP. The temperature profiles are largely similar between the GLAC-1D and ICE-6G_C runs, except between 26 and 20 ka BP, where the ICE-6G_C run gives temperatures consistently 5°C higher."

Section 4.1.1: The authors could refer to Fig. 2A in this part of the manuscript.

We have added references to Figure 2A (lines 182-188).

Section 4.1.2: The authors could provide the figures depicting the different appearances of Horizon Z (e.g. "... coincident with the seabed (profile A-A' in Fig. 5)."

We have added further figure references to show the character of Horizon Z in figures 1, 3, and 5 (lines 190-196).

Section 4.1.3: Similarly to the previous comment, the authors could provide the figures depicting the different types of appearances of the channel fill on the seismic sections (for example, for the acoustic blanking). Additionally, it would be beneficial for the readers, if the authors specify more in detail, where different details of the acoustic facies can be observed. The description between L171-172 could be "(middle part between 38 and 28 ms on profile A-A' in Figure 4)" instead of just "Figure 4". The described details are sometimes difficult to find in the figures (for example, I don't see the prograding fill in Fig. 4 and I don't even know which profile to observe).

We have added references to specific parts of figures (e.g. 4B, 4C; lines 199-206) and improved annotation of the figures to draw attention to seismic facies discussed in the text.

Section 4.1.4: Again, I suggest the authors provide more in detail where the described features of the acoustic facies can be observed in the profiles.

We have added further references to figures and improved annotation on figures in this section (lines 236-238).

L189-190: "In the north of the study area, the largest elongate feature can be observed to incise through the channel-fill unit and into the basal seismic unit" I suggest the authors refer the reader to a figure with a map which demonstrates this.

We have added a part to Figure 9 that shows this, and added a reference to Figure 9 in the text here (line 247).

L293: Subdued by what process? I suggest the authors use a word that is more descriptive or also reflects the possible process (e.g. relief was eroded, compacted, leveled out...).

We agree that "subdued" implies a process, but we simply meant "it's quite flat". We have reworded the sentence to reflect this (lines 410-414): "The resulting landscape surface is likely to have been modified where the seabed and Horizon Z are coincident, and therefore reconstructing the original topographic template is challenging, although it is likely that the topography was low relief, as part of this land surface beyond the channels is planar (Figure 3)."

L305: A reference to a figure would be appropriate after "accretion".

We have added a reference to Figure 4C here (line 424), and improved annotation on that figure.

L307: Maybe "1st part of Figure 8"

We have added reference to stages 1 and 2 of Figure 8 here (line 426).

L309: Do the authors have any idea, why the widths are relatively narrow and constant? Is it possible, that they were previously confined by relief (which is not preserved?). Are there any indices in the geophysical datasets for this?

We are uncertain as to why they are relatively constant, but it is likely a topographic constraint. We have discussed this further in this section, and why it may be difficult to infer subsequent erosion in the present dataset (lines 429-434): "This relatively constant width implies the existence of a topographic constraint, such as the low-relief valleys (Figure 3D), with the possibility that these valleys were once deeper, and the surrounding higher topography has been subsequently removed through wave ravinement during marine transgression (Emery et al., 2019b). It is difficult to test whether significant erosion has taken place due to the lack of a stratigraphic datum to correlate within the proglacial lake sediments, and such a correlation would require high vertical and spatial resolution of stratigraphic detail from borehole logs and seismic data that are beyond the capability of this dataset."

L316: The authors want to demonstrate that Channel 2 was located in a valley and probably mistakenly refer to the isopach map (Fig. 2B) instead to the horizon-depth map (map in Fig. 3). Nevertheless, I am not really convinced from Fig. 3 that channel 2 is located in a valley as it seems to be located on a topographical high of Horizon Z (between -30 and -33 m).

We have added profiles to Figure 3 to show the valleys we refer to, as they are subtle and do not show well on the map in Figure 3. We have also updated the figure reference here to reflect this (line 453).

L343: On which profile and where specifically is cross-bedding visible in Fig. 4?

Figure 4C. We have updated the figure reference here (line 479) and annotated figure 4C to reflect this.

L377-378: The authors state that the warmer-climate drainage network is best developed over the proglacial lake-fill sediments, however, the largest feature (Channel 3) is developed outside the bounds of the proglacial lake.

We have extended the sentence to include this observation (lines 522-524): "This in turn led to the development of the sub-dendritic drainage network, which is most developed and best preserved over the proglacial lake-fill sediments (Figure 6), except for main channel 3, which developed over basal sub-unit 1, which are glaciotectonised and overconsolidated clays."

L396-397: "sigmoidal to oblique reflectors in the upper seismic unit" - a reference to a figure would be appropriate in this part of the manuscript. "infill of the channels and the tidal scour features" – a figure showing a profile across the proposed tidal scours would be appropriate in this part of the manuscript.

We have added a reference to Figure 4C, and improved annotation of that figure, for the sigmoidal to oblique reflections (line 552), and added a part to figure 9 that shows the tidal scours, and added a reference to Figure 9 in the text (line 550).

L487-489: "Palaeoclimate modelling showed a cold, arid period between ice sheet retreat at 23 ka BP and 17 ka BP, when the climate became increasingly warm and wet, which correlates to marsh environments at Dogger Bank c. 14.9 – 13.5 ka BP." – Correlates in what way? A part of the sentence seems to be missing, as a cold period is regarded as a warm period. Or should "when" be "then"?; maybe "during ice sheet retreat between" instead of "between ice sheet retreat at"

The climate warming comes after the cold period, so we have reworded this sentence to reflect this climatic change (lines 675-677): "Palaeoclimate modelling showed a cold, arid period between ice sheet retreat at 23 ka BP and 17 ka BP, after which the climate became increasingly warm and wet, which correlates to marsh environments at Dogger Bank c. 14.9 – 13.5 ka BP."

L772-773: As this report is cited very often and is available online, I suggest the authors add the hyperlink to the report, if the journal guidelines allow.

We have added a hyperlink to the reference for this report (line 968).

Fig. 1: The font for the scale bar is disproportionally large compared to the other text on the figure. I also suggest to put the text for the depth and elevation colourbars below the colorbars. In that way, both texts are physically separated from the Forewind and Study area part of the legend and the legend becomes clearer. But these are just my personal preferences...

We have made these changes to Figure 1 to improve legibility.

Maps in Fig. 4, 5 and 9: It would be really beneficial for the readers to have the location of Figs. 4, 5 and 9 marked on one of the smaller scale Figures (Fig. 2B or Fig. 6 or...).

We have added the locations in Figures 4, 5 and 9 to Figure 3.

Fig. 3: What is the "m OD" abbreviation on the Horizon Z depth map?

Metres relative to Ordnance Datum. We have removed this reference to avoid confusion to international readers.

L786: Personally, I really appreciate the authors using "scientific colours" and hope others will follow.

Thank you! We do too.

Fig. 9: In L100 the authors mention that the reflections can be recognised up to 150 m deep. However, according to Fig. 9, the tunnel valley is more than 200 meters deep. If this is a mistake, the authors should correct the figure, otherwise I suggest you also include a reference to a previous study or a figure with a profile showing the tunnel valley (possibly 2 profiles to show the relation of the uneroded and eroded channel with the valley).

We have re-annotated the key of Figure 9 to make it clear this is depth below seabed, and the tunnel valley starts at around -66 m and extends to -226 m. We have also added a cross-section through the tunnel valley to show its relationship to main channel 1.

Fig. 11: The location of Oyster Ground is not marked on the map

We added an annotation to Figure 11 to show its location.

Technical corrections

L18: probably "represent a terrestrial" instead of "represent terrestrial"?

Yes, we have added "a" to this sentence (line 18)

L19: "comprises" instead of "comprise"

Corrected (line 19).

L28: probably "9 ka BP" instead of "8 ka BP"

More likely around 8.5-8 ka BP, this change is made here (line 28) and throughout the text to reflect this.

L108: maybe "and the extended interpretation of" instead of "and extended for interpretation of "

We have reworded this sentence for clarity (lines 120-121): "Seismic facies were identified and named based on Mitchum et al. (1977), with interpretation of glacial sediments using terminology based on Emery et al. (2019a)."

L114: "proxy for" instead of "proxy to"; alternatively you could use grain-size proxy

We have reworded this sentence for clarity and in line with comments made by another reviewer (lines 127-128): "These tests provide cone resistance (qc) measurements that were used, uncorrected, as a grain-size proxy through the sediments, with low resistance corresponding to clay and high resistance corresponding to sand, as used by Emery et al. (2019a)."

L119: "truncated. Using" instead of "truncated, using"

We have removed the second part of the initial sentence as it did not make sense and in line with comments by another reviewer.

L151: probably "Generally the area" instead of "Generally the"?

Yes, added (line 183).

L160-161: "Figure 2" should be "Figure 2b"?

Yes, added (line 191).

L186: Is "Figure 2" appropriate here? As "(Figure 2)" is placed at the end of the sentence, it seems that the authors are referring to a seismic profile and not an isopach map. If they are indeed referring to the map, I suggest they put "(Figure 2B)" after "thickest" in L185.

This sentence has been reworded to put "Figure 2B" after "thickest" (lines 242-243): "In central and northern parts of the study area, where the upper seismic unit is thickest (Figure 2B), low-frequency, low-amplitude, west to southwest-dipping sigmoidal to tangential oblique and shingled reflections are present."